# Single-cell RNA-seq reveals transcriptomic heterogeneity mediated by host–pathogen dynamics in lymphoblastoid cell lines

Elliott D SoRelle[1,2], Joanne Dai[1†], Emmanuela N Bonglack[1,3], Emma M Heckenberg[1], Jeffrey Y Zhou[4], Stephanie N Giamberardino[5], Jeffrey A Bailey[6], Simon G Gregory[5], Cliburn Chan[2], Micah A Luftig[1]*

[1]Department of Molecular Genetics and Microbiology, Center for Virology, Duke University School of Medicine, Durham, United States; [2]Department of Biostatistics and Bioinformatics, Duke University School of Medicine, Durham, United States; [3]Department of Pharmacology and Cancer Biology, Duke University School of Medicine, Durham, United States; [4]Department of Medicine, University of Massachusetts Medical School, Worcester, United States; [5]Duke Molecular Physiology Institute and Department of Neurology, Duke University School of Medicine, Durham, United States; [6]Department of Pathology and Laboratory Medicine, Warren Alpert Medical School, Brown University, Providence, United States

*For correspondence:
micah.luftig@duke.edu

Present address: [†]Amgen Inc, San Francisco, United States

**Abstract** Lymphoblastoid cell lines (LCLs) are generated by transforming primary B cells with Epstein–Barr virus (EBV) and are used extensively as model systems in viral oncology, immunology, and human genetics research. In this study, we characterized single-cell transcriptomic profiles of five LCLs and present a simple discrete-time simulation to explore the influence of stochasticity on LCL clonal evolution. Single-cell RNA sequencing (scRNA-seq) revealed substantial phenotypic heterogeneity within and across LCLs with respect to immunoglobulin isotype; virus-modulated host pathways involved in survival, activation, and differentiation; viral replication state; and oxidative stress. This heterogeneity is likely attributable to intrinsic variance in primary B cells and host–pathogen dynamics. Stochastic simulations demonstrate that initial primary cell heterogeneity, random sampling, time in culture, and even mild differences in phenotype-specific fitness can contribute substantially to dynamic diversity in populations of nominally clonal cells.

## Introduction

Lymphoblastoid cell lines (LCLs) are immortalized cells prepared by in vitro transformation of resting primary B cells from peripheral blood with Epstein–Barr virus (EBV) (*Bird et al., 1981*; *Anderson and Gusella, 1984*). LCLs are used extensively in research as a model for EBV-associated malignancies including diffuse large B cell lymphoma (*Nichele et al., 2012*; *Tazzari et al., 1999*) and post-transplant lymphoproliferative disorder (*Markasz et al., 2009*; *Rea et al., 1994*). Because EBV is a non-mutagenic transformant in this context, LCLs constitute an important renewable source of human cells and genomic DNA that are used in immunological, genetic, and virology research (*Çalışkan et al., 2014*; *Choy et al., 2008*; *Oh et al., 2013*; *Stark et al., 2010*; *Volkova et al., 2019*).

EBV is a double-stranded oncogenic gammaherpesvirus infecting over 90% of humans (*Rickinson and Kieff, 2007*). In vivo, the virus typically establishes an asymptomatic persistent latent

infection in episomal form (*Lindahl et al., 1976*; *Nonoyama and Pagano, 1972*) within resting memory B cells (*Longnecker et al., 2013*). Latent infection can take one of several forms, each characterized by distinct programs of viral gene expression initiated from different promoters (*Price and Luftig, 2015*). For example, classical EBV infection within resting memory B cells in vivo is characterized by the Latency I program in which expression from the Q promoter yields a single viral protein, EBV Nuclear Antigen 1 (EBNA1), which functions to maintain the viral episome (*Hung et al., 2001*). Latency I, termed 'true latency', is established only after a complex progression of infection through pre-latency, Latency IIb, Latency III, and restricted forms of latency (e.g., Latency IIa), each occurring in distinct tissues within the body (*Price and Luftig, 2015*). EBV can undergo lytic reactivation as a replication strategy, which is relatively infrequent in cell culture despite being essential for transmission in vivo (*Bhende et al., 2004*).

In vitro, the process of LCL production also necessarily involves multiple transitions in viral transcriptional programs. In the immediate-early stage of infection (the pre-latent phase), expression from the W promoter yields EBNA-LP, EBNA2, and several noncoding RNAs (EBERs, BHRF1 miRNAs, and BART miRNAs). A brief burst of lytic gene transcription (without lytic replication) is also observed during pre-latency (*Woisetschlaeger et al., 1989*). EBNA-LP and EBNA2 protein levels increase gradually within these early-infected cells, eventually leading to Latency IIb in which EBNA2 activation of the C promoter upregulates expression of EBNA1, EBNA3A, EBNA3B, EBNA3C, and additional EBNA-LP and EBNA2 proteins as well as noncoding RNAs (*Alfieri et al., 1991*). Latency IIb gene products induce hyperproliferation, a period of several days during which infected B cells divide every 10–12 hr (*Nikitin et al., 2010*). During hyperproliferation, EBNA1 mediates viral genome replication while EBNA3 proteins inhibit host cell antiviral and tumor-suppression responses. Variance in virally mediated rates of proliferation ensures that some infected cells undergo DNA damage-induced growth arrest (*Nikitin et al., 2010*; *Nikitin et al., 2014*) while others continue to proliferate, eventually outgrowing as immortalized LCLs. LCLs largely exhibit the Latency III transcriptional profile, characterized by expression of all six EBNAs (EBNA-LP, EBNA1, EBNA2, and EBNAs 3A–3C) in addition to latent membrane proteins 1 and 2 (LMP-1, LMP-2A/B) and noncoding RNAs (*Young and Rickinson, 2004*). In Latency III, EBNA2 stimulates expression of LMP-1, a constitutively active tumor necrosis factor receptor (TNFR) homolog (*Mosialos et al., 1995*). LMP-1 signaling drives proliferation and survival via NFκB pathway activation (*Devergne et al., 1996*), which has been shown to be essential for LCL outgrowth (*Cahir-McFarland et al., 2000*).

Although studied extensively, complete characterization of the viral and host determinants of growth arrest versus immortalization of early-infected cells remains elusive (*Mrozek-Gorska et al., 2019*). As one consequence, it is unclear whether or to what extent viral transformation may influence the resulting LCL cell populations. The possibility of significant phenotypic diversity within and across LCL samples warrants consideration, given the intrinsic variance of the human primary B cell repertoire (*Morbach et al., 2010*; *Perez-Andres et al., 2010*) and the multiplicity of viral transcription programs active in the journey to immortalization. Indeed, we recently described a gene expression program having low expression of LMP1 and NFκB targets which was unique to early infection (Latency IIb) relative to an otherwise identical population of LCLs (*Messinger et al., 2019*). The wide distribution in LMP1 and NFκB target expression levels within an LCL has been characterized and ascribed to the dynamic sampling of a distribution of immune evasive states, at the fringes of which growth and survival can be compromised (*Brooks et al., 2009*; *Lam et al., 2004*; *Lee and Sugden, 2008*).

In this study, we characterize the transcriptomic profiles of five different LCLs with single-cell resolution to assess inter- and intra-sample heterogeneity. Four of the sampled LCLs (two in-house and two commercial cell lines) were transformed with the prototypical B95-8 strain of EBV derived from an infectious mononucleosis patient (*Miller and Lipman, 1973*), while a fifth sample (in-house) was prepared from cells transformed with the M81 strain isolated from a human nasopharyngeal carcinoma sample (*Desgranges et al., 1976*; *Tsai et al., 2013*). Primary cells used in establishing the five LCLs were isolated and transformed from a total of four donors; cells from one donor were transformed concomitantly to establish LCLs with each of the tested EBV strains. We observe B cell genetic heterogeneity in the form of differential heavy chain isotype expression across LCLs and, in three instances, within a sample. Further, comparable patterns of phenotypic variance with respect to NFκB pathway and plasma cell-like differentiation genes are evident in each LCL. Expression of host and viral genes indicate that individual cells within LCLs occupy a continuum of infection states.

We also present an initial stochastic model to explore factors beyond the nuances of host–pathogen interactions that may generate profound phenotypic diversity within cultured cell lines. Our findings highlight some of the underappreciated complexity inherent within LCLs and broadly underscore the importance of understanding and accounting for sources of heterogeneity within presumptive cell lines.

## Results

### LCL generation and data provenance

Three LCLs were prepared in-house by infection of PBMCs from two donors (sample numbers 461 and 777) with one of two different EBV strains (B95-8 or M81). Each of these three samples (LCL 461 B95-8, LCL 777 B95-8, and LCL 777 M81) was prepared and processed using standard single-cell RNA sequencing workflows (see Materials and methods). Two additional, publicly available data sets were obtained for commercially available samples of the GM12878 and GM18502 LCLs, which were generated as previously reported by Osorio and colleagues (*Osorio et al., 2019*). These five samples yielded single-cell RNA count matrices for subsequent analysis.

### LCL sample quality control (QC)

Count matrices for the five samples exhibited similar feature, total RNA count, and mitochondrial gene distributions (*Figure 1—figure supplements 1* and *2*) and were subjected to standardized QC thresholding (see Materials and methods). Cell cycle marker expression (*Figure 1—figure supplement 3*) was scored and regressed out during selection of highly variable genes as features to avoid clusters arising solely from cell cycle phase. Selected features were used to derive principal components which were evaluated (*Figure 1—figure supplement 4*) and subsequently used for dimensional reduction (see Materials and methods). Separate analysis of the merged sample data set indicated that inter-donor variability is the predominant source of heterogeneity (*Figure 1—figure supplement 5*).

### Immunoglobulin isotype heterogeneity within and across LCL samples

The five LCL populations exhibit distinct immunoglobulin (Ig) profiles with respect to both gene expression levels and isotype frequencies (*Figure 1*). Three of the five samples (LCL 777 B95-8, LCL 777 M81, and GM12878) contain IgM$^+$ and class-switched IgA$^+$ and IgG$^+$ subpopulations, whereas two samples (LCL 461 B95-8 and GM18502) almost exclusively expressed IgG (*IGHG1-4*; *Figure 1A*). Additionally, cells within each isotype class exhibit a wide range of Ig transcript levels across all samples in an apparent class-independent fashion. No significant expression of *IGHE* was observed in any of the five samples, consistent with the isotype's rarity in the peripheral blood (*He et al., 2017*; *Saunders et al., 2019*). The immunoglobulin compositions observed for each LCL were confirmed subsequently by RT-PCR and sequencing, which revealed that each isotype represents a distinct clone within the culture (*Figure 1—figure supplement 6*). Significant *IGHD* transcript levels were observed in one sample (LCL 777 B95-8), where the gene's expression was constrained to (and varied inversely with expression levels of) IgM$^+$ cells (*Figure 1—figure supplement 7*).

The proportion of cells expressing each isotype varied substantially among LCLs (*Figure 1B*). IgG was the only isotype observed in LCL 461 B95-8. Cells in the GM18502 sample were also homogenous for *IGHG1*, although low levels of *IGHM* transcripts are observed in up to half of the population. The proportion of IgM$^+$, IgA$^+$, and IgG$^+$ subpopulations in LCL 777 B95-8 were 69%, 7%, and 24%; in LCL 777 M81 were 1%, 35%, and 64%; and in GM12878 were 6%, 73%, and 18%. Abundance of Ig light chain gene (kappa or lambda) and heavy chain isoform expression are generally correlated with variable heavy chain expression in each of the five samples (*Figure 1—figure supplements 7–16*). The isotype and clonal frequency differences between LCL 777 B95-8 and LCL 777 M81 are notable, given that these samples originated from the same donor and were transformed at the same time with different viral strains.

Differential Ig isotype expression is a significant source of variation in LCLs, as captured by the loadings from principal component analysis (PCA), typically within the first four PCs. Consequently, differences in Ig isotype are effectively captured in dimensionally reduced data sets generated from PCs using t-distributed stochastic neighbor embedding (tSNE) even at low clustering resolution. In

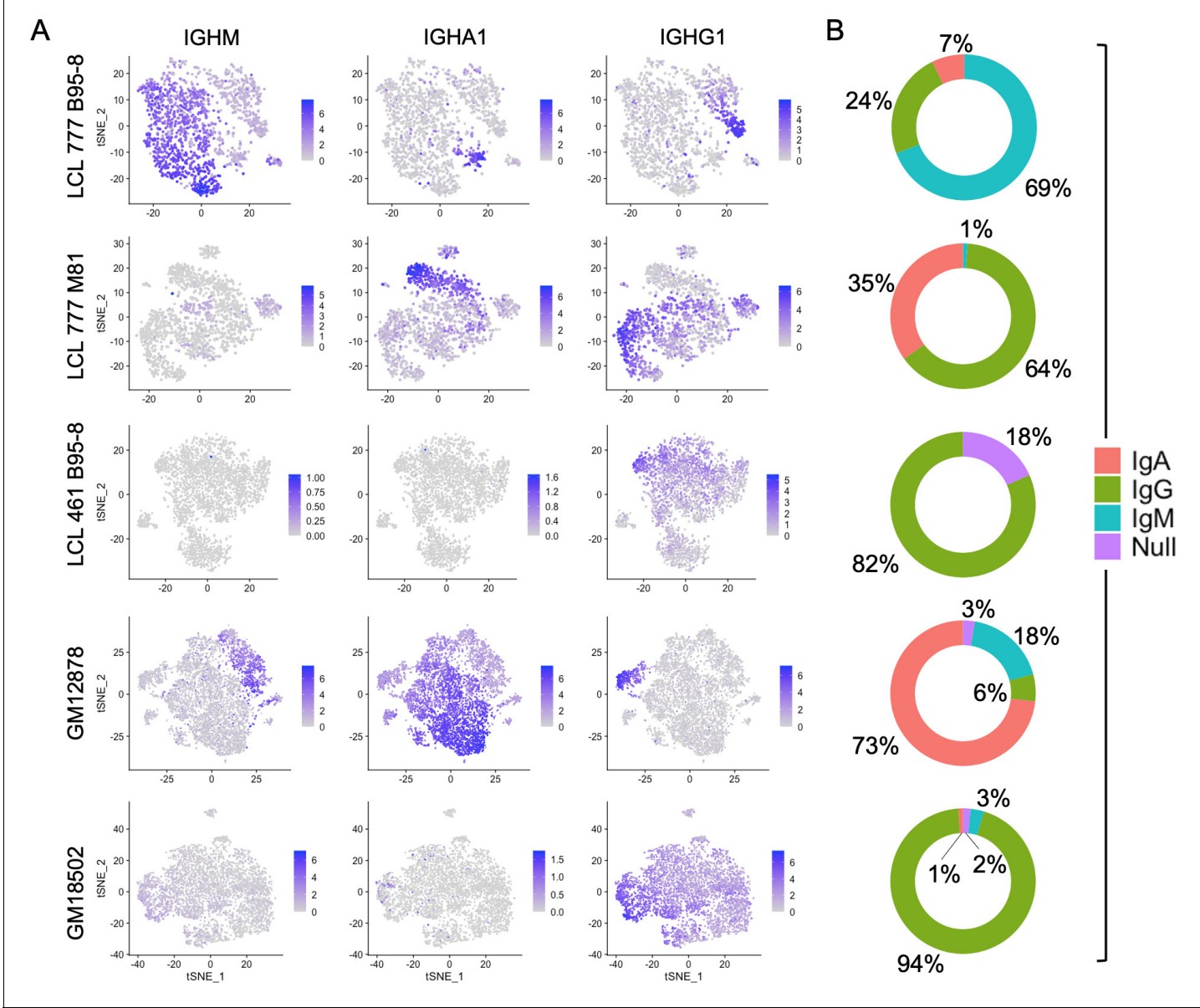

**Figure 1.** Immunoglobulin isotype heterogeneity within and across lymphoblastoid cell lines (LCLs). (**A**) Relative expression of immunoglobulin heavy chain genes (*IGHM*, *IGHA1*, and *IGHG1*) in five LCLs analyzed by single-cell RNA sequencing. Data are represented by dimensional reduction (t-distributed stochastic neighbor embedding) of principal components generated from feature selection following out-regression of cell cycle markers (see Experimental methods). (**B**) Percentage of cells in LCL population within each isotype class. Null classification represents cells exhibiting negligible immunoglobulin heavy chain expression.

The online version of this article includes the following figure supplement(s) for figure 1:

**Figure supplement 1.** Distributions of features used for QC across five lymphoblastoid cell line (LCL) samples.
**Figure supplement 2.** Summary of QC statistics across five lymphoblastoid cell line (LCL) samples.
**Figure supplement 3.** Distributions of representative markers used for cell cycle scoring and regression.
**Figure supplement 4.** Elbow and Jackstraw plots used for determination of principal components to use for dimensional reduction and clustering.
**Figure supplement 5.** Merged sample analysis.
**Figure supplement 6.** Validation of Ig heavy and light chain (poly)clonality for five lymphoblastoid cell lines (LCLs).
**Figure supplement 7.** Expression of key gene groups in LCL 777 B95-8.
**Figure supplement 8.** Expression of key gene groups in LCL 777 M81.
**Figure supplement 9.** Expression of key gene groups in LCL 461 B95-8.
**Figure supplement 10.** Expression of key gene groups in GM12878.
**Figure supplement 11.** Expression of key gene groups in GM18502.

*Figure 1 continued*

**Figure supplement 12.** Pairwise Pearson correlation values across key gene groups in LCL 777 B95-8.
**Figure supplement 13.** Pairwise Pearson correlation values across key gene groups in LCL 777 M81.
**Figure supplement 14.** Pairwise Pearson correlation values across key gene groups in LCL 461 B95-8.
**Figure supplement 15.** Pairwise Pearson correlation values across key gene groups in GM12878.
**Figure supplement 16.** Pairwise Pearson correlation values across key gene groups in GM18502.

samples with more homogenous isotype expression (LCL 461 B95-8 and GM18502), the relative Ig expression level is a significant factor in distinguishing clusters.

## Genes involved in B cell activation and differentiation exhibit inverse expression gradients

Across all samples, LCL populations display variable mRNA transcript levels for genes involved in cell activation, inhibition of apoptosis, response to oxidative stress, and differentiation (*Figure 2*). Gradients in Ig expression exhibit strong anticorrelation with expression of NFκB pathway transcripts (e.g., *NFKB2*, *NFKBIA*, and *EBI3*) central to B cell activation and survival (*Figure 2A*, *Figure 2—figure supplement 1A*). Similar gradients are observed for metabolic and oxidative stress response transcripts (e.g., *TXN*, *PRDX1*, *PKM*, *LDHA*, *ENO1*, and *HSP90AB1*); however, these transcripts are present more broadly (>80% of cells) and at higher levels than NFκB-related genes (20–30% of cells) in each sample (*Figure 2—figure supplement 2*). While NFκB family gene expression is consistently anticorrelated with that of B cell differentiation factors, significant diversity exists in NFκB-high cells with respect to specific subunits including *REL*, *RELA*, and *RELB* (*Figure 2—figure supplement 3*). This implies differential intercellular NFκB dimer composition and, consequently, intra-sample variation in NFκB-mediated transcriptional programs. Expression of NFκB regulated BCL2 family members (e.g., *BCL2L1*/Bcl-xL and *BCL2A1*/BFL1) displays strong anticorrelation with Ig expression level. However, *MCL1* and *BCL2* mRNAs are more broadly expressed across cells within each LCL, while *BCL2L2*/BCL-W is only modestly expressed in LCLs (*Figure 2—figure supplement 4*).

Ig gradients are closely related to expression of differentiation and maturation markers (e.g., *CD27*, *TNFRSF17*/BCMA, *XBP1*, *MZB1*, and *PRDM1*) (*Bhende et al., 2007*; *Hatzoglou et al., 2000*; *Rosenbaum et al., 2014*), which are likewise anticorrelated with NFκB pathway markers (*Figure 2B and C*, *Figure 2—figure supplement 1B*). The apparent inverse relationship between these gene sets defines a major axis of phenotypic variance within LCL samples comprising multiple Ig isotypes (*Figure 2D*). The orthogonality of the pro-survival/differentiation and isotype class diversity axes implies that these two aspects of phenotypic variance are decoupled. Continuity between phenotypes resembling activated B cells (ABC) and antibody-secreting cells (ASC) is also captured in the expression profiles of key genes involved in the mutually antagonistic control of B cell state (*Figure 2—figure supplement 5*; *Nutt et al., 2015*). In this model, genes including *PAX5* and *IRF8* promote the ABC state; *IRF4* and *MKI67* (a G2/M cell cycle marker) are markers of a transitional phenotype; and *PRDM1* (BLIMP1) and *XBP1* promote the ASC state. As cell cycle marker expression was regressed out, mitotic phase has negligible influence on the observed trends.

Whereas distinctions in Ig isotype class expression tend toward discrete partitioning, intra-isotype expression of differentiation and maturation genes reflects a continuum of transcriptomic states and cellular functions. Thus, within a given isotype, elevated Ig heavy chain expression is negatively correlated with activation/anti-apoptotic gene expression and positively correlated with maturation/differentiation gene expression. These relationships are most readily evident in LCL samples consisting of a single class-switched population, such as GM18502 (*Figure 2—figure supplement 1C*).

Finally, the viral EBNA2 and EBNA3 proteins are responsible for transcriptional regulation that we specifically interrogated within the single cell data. The direct EBNA2 targets *RUNX3* and *FCER2*/CD23 correlated with NFκB expression (*Figure 2—figure supplement 6*; *Spender et al., 2002*). Indeed, the expression of *RUNX3* and *FCER2*/CD23 was anticorrelated with Ig expression consistent with the known role of EBNA2 in suppressing heavy chain transcription (*Jochner et al., 1996*). In contrast, the EBNA3 repressed targets including *CXCL9*, *CXCL10*, *BCL2L11*/BIM, and *ADAMDEC1* were uniformly repressed (*Figure 2—figure supplement 7*) consistent with the role of histone and

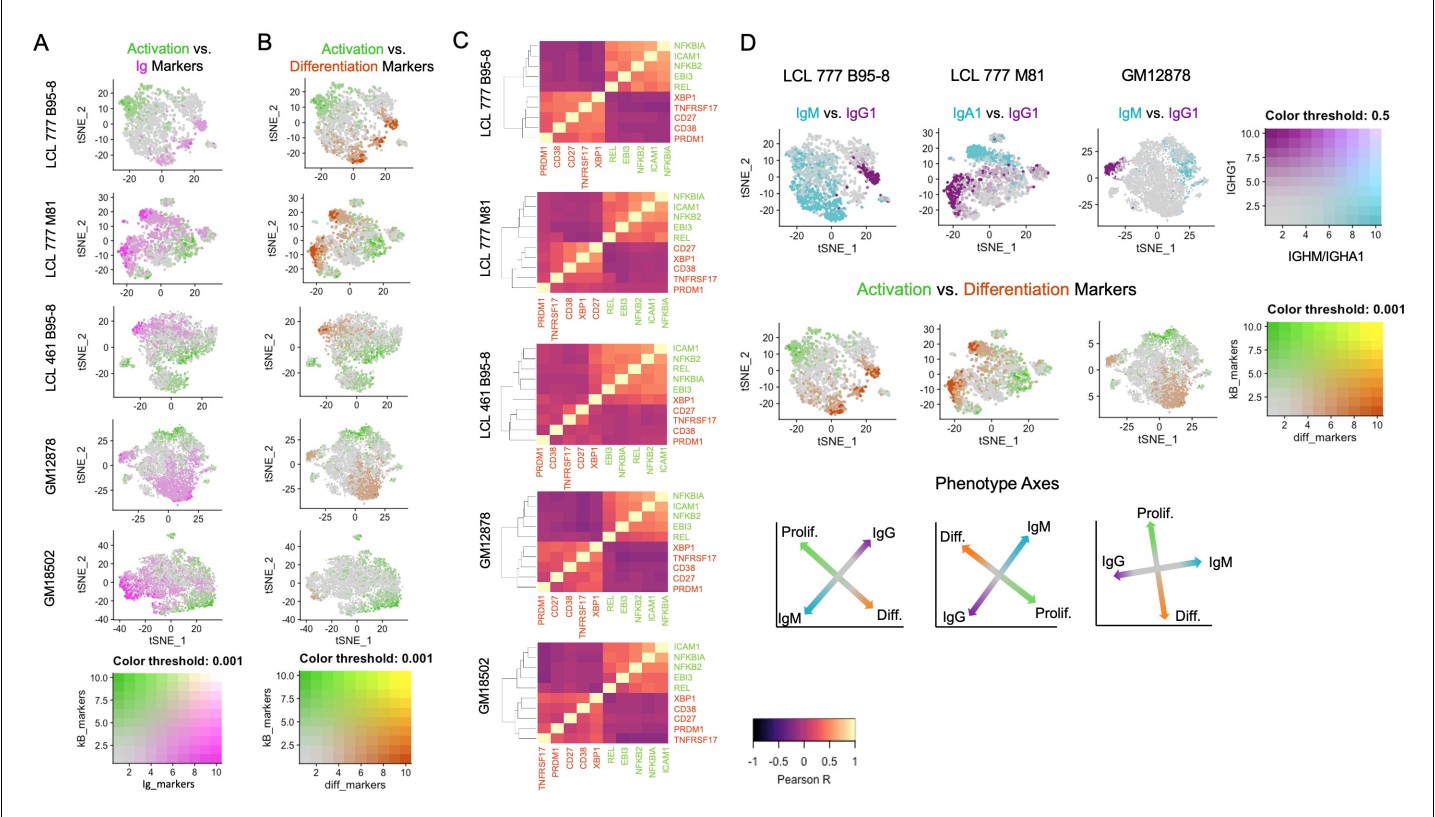

**Figure 2.** Lymphoblastoid cell lines (LCLs) exhibit anticorrelated expression gradients of activation and differentiation genes. (**A**) Inverse expression gradients of immunoglobulin genes (*IGHM*, *IGHA1*, and *IGHG1*) in magenta and NFκB targets (*NFKB2*, *NFKBIA*, *EBI3*, *ICAM1*, and *BCL2A1*) and *TXN* in green. (**B**) Similar inverse gradients of NFκB targets in green and B cell differentiation markers (*TNFRSF17*, *XBP1*, *MZB1*, *CD27*, and *CD38*) in orange. (**C**) Pearson correlation maps and hierarchical clustering reveal negative correlation of differentiation (orange) and activation (green) gene sets and positive correlations between genes within each set. (**D**) In LCLs comprising multiple immunoglobulin isotypes, heavy chain class and differentiation/activation gradients constitute orthogonal (independent) axes of phenotypic variance.

The online version of this article includes the following figure supplement(s) for figure 2:

**Figure supplement 1.** Expression of individual genes within activation and differentiation gene sets.

**Figure supplement 2.** Expression of metabolic and oxidative stress genes.

**Figure supplement 3.** Expression of NF-κB subunits c-REL, RELA, and RELB.

**Figure supplement 4.** Expression of BCL2 family genes across lymphoblastoid cell line (LCL) samples.

**Figure supplement 5.** Expression trends in key transcriptional regulators controlling activated B cell (ABC) and antibody-secreting cell (ASC) phenotypes.

**Figure supplement 6.** Expression of host targets upregulated by EBNA2.

**Figure supplement 7.** Expression of host targets repressed by EBNA3.

**Figure supplement 8.** Cell proliferation and metabolic profiling by ICAM expression.

DNA methylation in maintaining gene repression of EBNA3 targets (*Harth-Hertle et al., 2013*; *McClellan et al., 2013*; *Paschos et al., 2009*).

Differential expression of genes involved in cell activation could affect rates of cell proliferation within an LCL population. To explore this possibility, we sorted three additional LCLs by ICAM-1 expression and evaluated the growth and metabolic profiles of the sorted fractions. On average, ICAM-1[hi] cells (consistent with the ABC phenotype) exhibited modestly faster growth in culture than ICAM-1[lo] cells (ASC phenotype) between 1 and 4 days post-sorting. Notably, metabolic activity was elevated in ICAM-1[hi] cells than ICAM-1[lo] cells across all three LCLs, as indicated by higher rates of glycolysis and oxygen consumption (*Figure 2—figure supplement 8*).

## Viral state heterogeneity affects host expression profile distributions in LCLs

Clusters with high EBV lytic gene expression are observed in two of the three data sets (LCL 777 B95-8 and LCL 777 M81) aligned against the human reference genome containing the viral genome as an extra chromosome (see Materials and methods; *Figure 3*). Lytic cluster cells are small, accounting for 2.2% and 0.9% of the LCL 777 B95-8 and LCL 777 M81 cell populations, respectively (*Figure 3A*). The higher rate of lytic cell capture in the B95-8 sample relative to the M81 sample is somewhat surprising, as the M81 strain is known for increased frequency of lytic reactivation; however, this disparity may originate from the nature of single-cell sample preparation method (see Discussion; *Zheng et al., 2017*).

The presence or absence of viral lytic transcripts is a significant source of phenotypic variance in these samples, as reflected in population groupings by viral state (*Figure 3B*) and principal component loadings (*Figure 1—figure supplements 15* and *16*, PC_3 and PC_7, respectively). Lytic cells can be identified confidently from high expression of EBV genes including *BLRF1*, *BALF1*, and *BARF1*, among others (*Figure 3C*). *BHRF1* expression is also elevated in lytic cells, although *BHRF1* transcripts are ubiquitous at low levels sample wide. This is likely because *BHRF1* can be expressed during both latent and lytic phases of EBV infection from different promoters (*Xing and Kieff, 2007*). Cells identified as lytic exhibit lytic gene expression ranging from approximately 3–15% of total measured transcripts per cell (*Figure 3—figure supplement 1*). Thus, it is possible that this cluster represents both truly lytic cells (>10% lytic transcripts) and abortive lytic cells (*Chiu and Sugden, 2016*). Alternatively, the cells with lower lytic transcript expression may have been at earlier stages of lytic reactivation at the time of sample preparation.

While the absolute number of lytic cells in each sample is low, the data indicate that the lytic cells are polyclonal with respect to Ig heavy chain expression, display upregulation of several host genes including *NFATC1*, *MIER2*, *SFN*, and *SGK1*, and exhibit heterogeneous NFκB expression

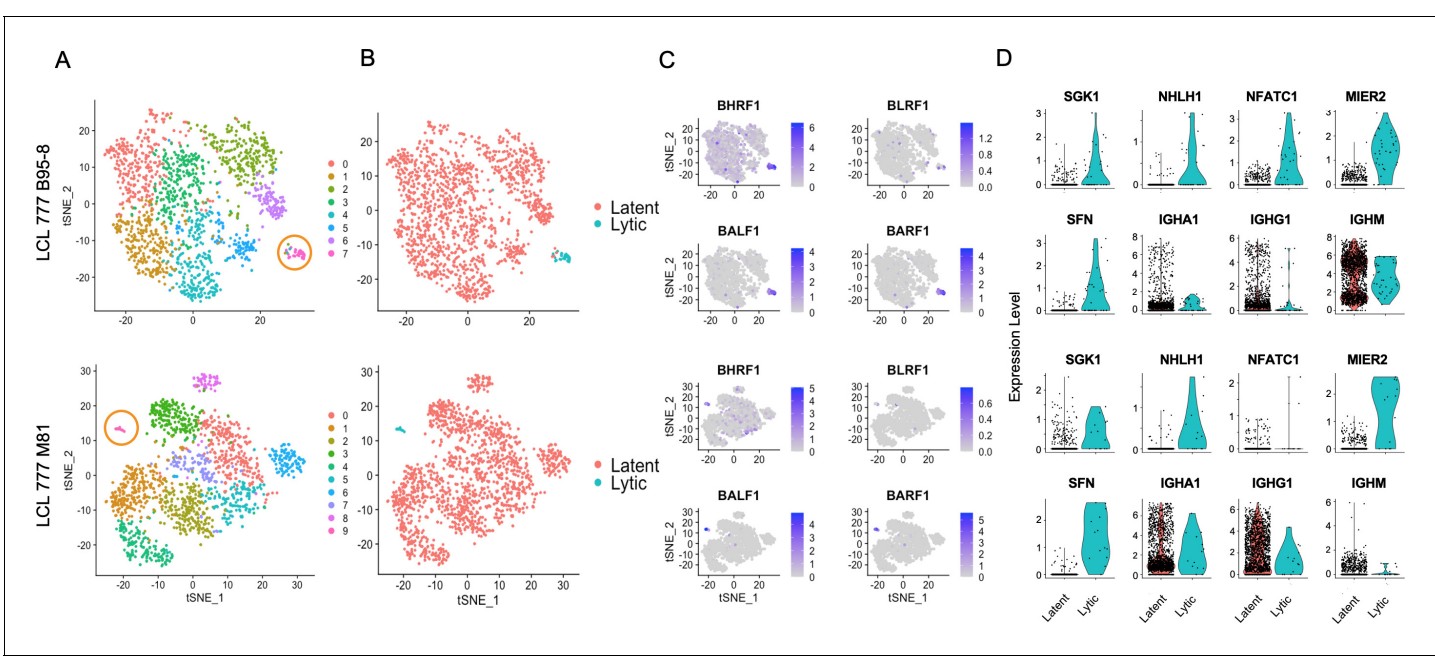

**Figure 3.** Viral and host gene expression in lytic cell subpopulations. (A) Clustering of dimensionally reduced data sets for LCL 777 B95-8 and LCL 777 M81. (B) Grouping of cell clusters into latent (red) and lytic (cyan) cells based on viral and host gene expression signatures of principal components. (C) Relative expression of four representative Epstein–Barr virus (EBV) lytic genes (*BHRF1*, *BLRF1*, *BALF1*, and *BARF1*) is elevated in lytic cell subpopulations. (D) Lytic cell clusters exhibit elevated expression of several host cell genes (*SGK1*, *NHLH1*, *NFATC1*, *MIER2*, and *SFN*) relative to latently infected cells. While under-sampled due to subpopulation size, immunoglobulin class frequencies in lytic cells roughly reflect the population-wide frequencies.

The online version of this article includes the following figure supplement(s) for figure 3:

**Figure supplement 1.** Percentage of viral lytic transcripts relative to total transcripts in LCL_777_B95-8 lytic cell cluster.

(*Figure 3D*, *Figure 1—figure supplements 5* and *6*). Ig isotype distributions in lytic cell clusters appear roughly proportional to the whole-sample distributions. *NFATC1*, *MIER2*, *SFN*, and *SGK1* transcript levels were queried for GM12878 and GM18502 samples to test whether the presence of lytic cell subpopulations might be inferred from host gene expression. A sub-cluster representing a small percentage of cells in GM12878 (<0.5%) were found to co-express *MIER2* and *NFATC1*. Negligible expression of either gene was observed in GM18502 (*Figure 1—figure supplements 8* and *9*).

## Loss of mitochondrial and Ig expression in subpopulations under oxidative stress

Three of the five samples (LCL 461 B95-8, GM12878, and GM18502) contain clusters that exhibit metabolic transcriptional profiles in stark contrast with typical expression in each population (*Figure 4*). Cells within these clusters account for 1–4% of the three samples after QC (*Figure 4A*) and are most notable for their low expression of mitochondrial genes (*Figure 4B*). In the case of LCL 461

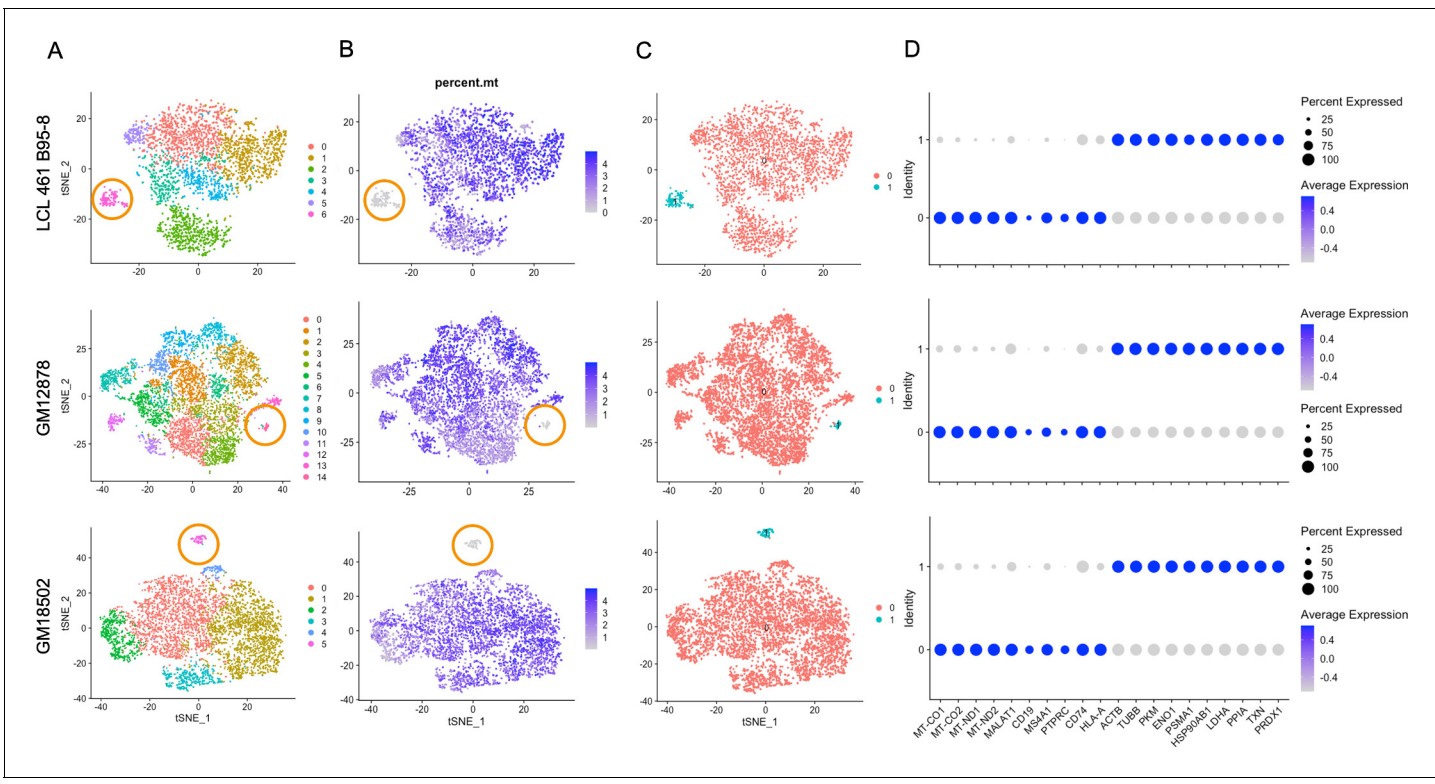

**Figure 4.** Lymphoblastoid cell line (LCL) subpopulations exhibiting reduced mitochondrial gene expression and elevated metabolic and oxidative stress genes. (**A**) Clustering of dimensionally reduced data sets for LCL 461 B95-8, GM12878, and GM18502. (**B**) Distinct clusters within each of these samples are defined by uncharacteristically low mitochondrial gene expression. (**C**) Grouping of cell clusters to partition 'mito-low' cells (cyan) for differential expression comparison. (**D**) Mito-low cells exhibit reduced expression of cytochrome oxidase (*MT-CO1* and *MT-CO2*), NADH-ubiquinone oxidoreductase (*MT-ND1* and *MT-ND2*), *MALAT1*, and numerous lymphoid and B-cell lineage markers (*CD19*, *MS4A1*/CD20, *PTPRC*/CD45, *CD74*, and *HLA-A*). Mito-low cells exhibit increased expression of genes associated with cytoskeletal rearrangements (*ACTB* and *TUBB*), metabolic stress (*PKM*, *ENO1*, and *LDHA*), protein folding/degradation (*HSP90AB1*, *PSMA1*, and *PPIA*), and oxidative stress (*TXN* and *PRDX1*).

The online version of this article includes the following figure supplement(s) for figure 4:

**Figure supplement 1.** Clustering resolution screens for LCL 777 B95-8.

**Figure supplement 2.** Clustering resolution screens for LCL 777 M81.

**Figure supplement 3.** Clustering resolution screens for LCL 461 B95-8.

**Figure supplement 4.** Clustering resolution screens for GM12878.

**Figure supplement 5.** Clustering resolution screens for GM18502.

**Figure supplement 6.** Expression of MHC class I genes HLA-A, HLA-B, and HLA-C.

**Figure supplement 7.** Total RNA counts, unique feature, and mitochondrial percentage distributions across lymphoblastoid cell line (LCL) samples.

B95-8 and GM18502, these cells are the first to partition from the rest of the sample at low clustering resolution (*Figure 4—figure supplements 1–5*).

Compared to the rest of each sample, these atypical cells exhibit significantly depleted levels of cytochrome c oxidase (*MT-CO1* and 2, complex IV) and NADH-ubiquinone oxidoreductase subunits (*MT-ND1 and 2*, complex I) as well as a lack of canonical markers of lymphoid (e.g., *PTPRC/CD45*, *CD74*), B cell-specific lineage (e.g., *CD19*, *MS4A1/CD20*), and in some cases, MHC class I and II antigen presentation (e.g., *HLA-A*, *HLA-B*, *HLA-C*, and *HLA-DR*; *Figure 4C and D*, *Figure 1—figure supplements 7* and *9*, *Figure 4—figure supplement 6*).

Expression of genes involved in oxidative stress (*TXN* and *PRDX1*), unfolded protein responses (*PPIA* and *HSP90AB1*), metabolic shunt pathways (*PKM*, *ENO1*, and *LDHA*), and cytoskeletal rearrangements (*ACTB* and *TUBB*) is enriched consistently in this subset relative to the bulk population in each of the three LCLs (*Figure 4D*, *Figure 2—figure supplement 2*). Ig heavy chain transcripts are notably absent from these subpopulations, although some degree of light-chain expression is observed (*Figure 1—figure supplements 7–9*). While these cells are on the low end of the population distribution with respect to total RNA counts and unique feature RNAs (*Figure 4—figure supplement 7*), the measured values are consistent with intact, viable cells.

## A stochastic model for LCL phenotypic heterogeneity

A simple stochastic simulation based on a discrete-time Markov chain model (*Škulj, 2006*) was developed to understand better the factors that may influence phenotypic heterogeneity observed in LCLs, using Ig isotype frequencies as an example (*Figure 5*). In principle, the simulation may be adapted to any set of phenotypes within a sample. For additional details regarding model parameters and assumptions, please see the Materials and methods (Stochastic simulations) and refer to the source code (*Source code 2*).

In the present implementation, changes in Ig isotype frequency can be simulated in discrete steps (rounds of cell division) as a function of initial phenotype frequencies, population sampling (with replacement), and potential differences in phenotypic fitness captured as fixed, (un)equal isotype-specific proliferation probabilities. The model assumes a fixed cell death rate across all isotypes in any given division round. The number of simulated trials can be adjusted to capture individual stochastic realizations or probabilistic outcome distributions. Each parameter and assumption can be adjusted by the user for tailored applications.

Three randomly selected realizations and averaged outcomes (trials = 100) of the model for a fixed sample size (n = 1000 cells) demonstrate the effects of intrinsic stochasticity on the evolution of phenotype proportions over many rounds of cell division (rounds = 300), even when each phenotype confers equivalent fitness (*Figure 5A*). In the case of equal fitness and sufficient sample size, initial phenotype frequencies are a key determinant of whether the most prevalent phenotype will change over time because of stochasticity.

The effect of sample size on inter-trial variance can be substantial, even when cell populations are sampled with replacement to maintain phenotype proportions in each round (*Figure 5B*). Mean phenotype proportions are generally conserved, whereas trial standard deviation decreases as the sample size increases (trials = 25, rounds = 300, n = 100, 500, 1000, or 5000 cells). This is generally expected, since undersampling increases the likelihood that phenotype frequencies in the drawn sample will deviate from those of the population, even in the case of replacement.

It is notable that minor differences in relative fitness (1–2%) can lead to dramatic changes in isotype distributions over time (*Figure 5C*). The rate of such change is proportional to the magnitude(s) of fitness differences (n = 1000 cells, rounds = 300). Four randomly selected clonal evolution trajectories realized with a modest fitness advantage (2%) for class-switched cells (IgA, IgE, and IgG) reveal the potential for drastic variations when multiple rare phenotypes with a fitness advantage exist (n = 2500 cells, trials = 10, rounds = 1000). Thus, rare cells may become prevalent or even dominant over time if they exhibit only slightly greater fitness relative to other cells in some environmental context (e.g., cell culture). In such cases, observed phenotype frequencies can deviate wildly from expectations of equal fitness over time (*Figure 5D*).

Cluster simulation was implemented by random sampling from four arbitrary, isotype-specific 2D normal distributions based on empirical observations that Ig isotypes yield distinct clusters in dimensionally reduced single-cell RNA-seq data (*Figure 5E–G*). Simulated clusters were generated from randomly selected trials initiated from the same initial phenotype distribution (IgM = 89%; IgA = 5%;

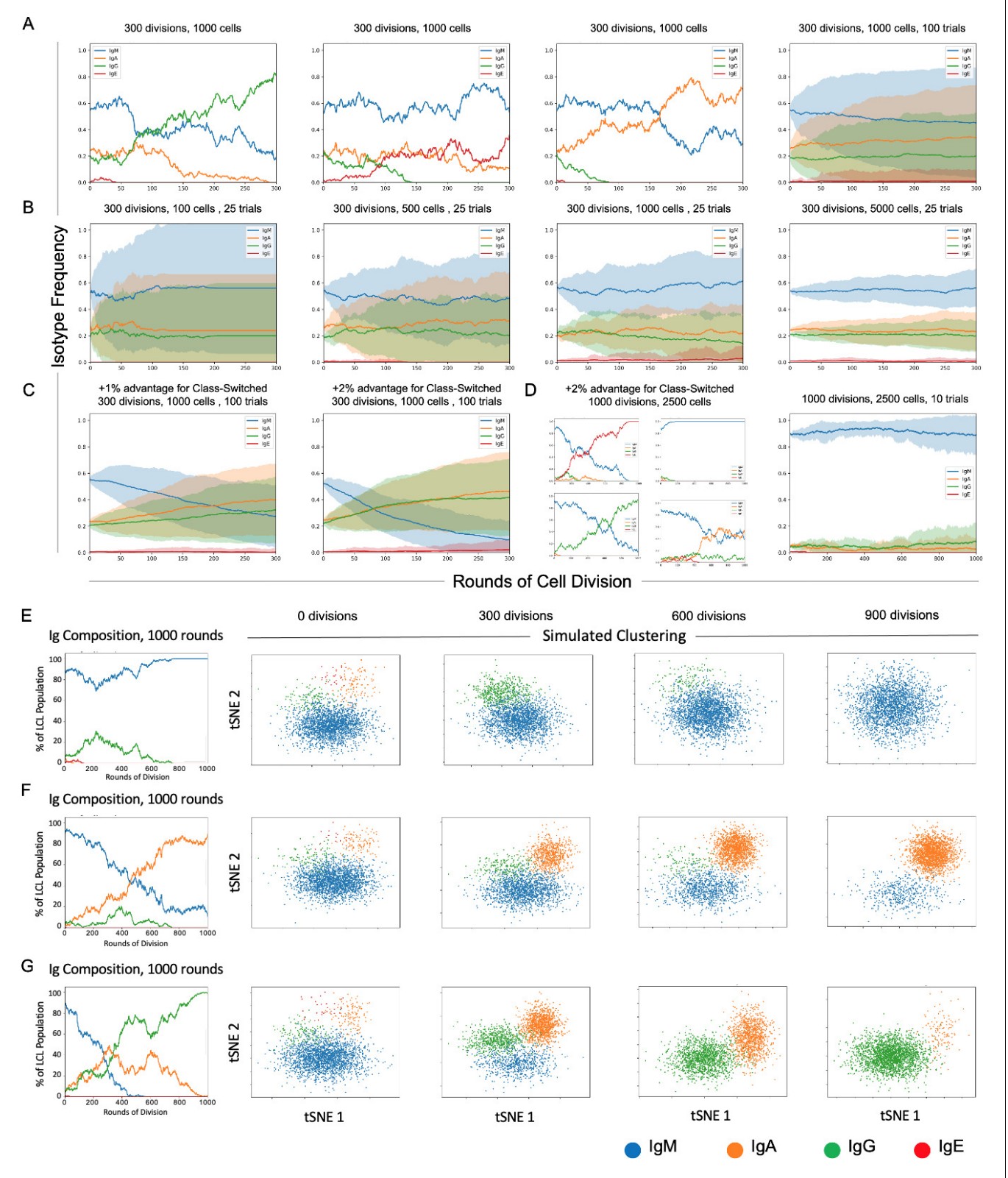

**Figure 5.** Stochastic simulation of heterogeneous lymphoblastoid cell line (LCL) evolution. (**A**) Stochastic immunoglobulin isotype frequency evolution. Three random single-trial simulations initiated from the same starting class frequencies are presented, assuming equal likelihood of proliferation across isotype classes (n = 1000 cells). The last panel shows mean and standard deviation for outcomes from 100 trials simulated from the same parameters. (**B**) Simulation of a founder effect. Population under-sampling (modeled by comparing results from 25 trials using n = 100, 500, 1000, and 5000 cells,

*Figure 5 continued on next page*

*Figure 5 continued*

left-to-right panels) increases outcome variance and accelerates convergence to a single isotype. (C) Effect of phenotype-specific fitness advantages. Simulation results are presented for scenarios in which class-switched isotypes (IgA, IgG, and IgE) have a 1% (left panel) or 2% (right panel) fitness advantage over IgM cells. (D) Four random single-trial simulations over long periods of time (1000 division rounds) with a 1% fitness advantage for class-switched cells (left panels) compared to 10 trials over the same period with equal fitness across classes. (E) Single-trial isotype frequency evolution and corresponding simulated clustering (see Materials and methods) in the case of equal proliferation probability. Starting frequencies of IgM, IgA, IgG, and IgE cells are 89%, 5%, 5%, and 1%, respectively. (F) As in E, with a 1% fitness advantage for class-switched cells. (G) As in E, with a 2% advantage for class-switched cells.

IgG = 5%; IgE = 1%) at three different relative fitness advantages (0%, 1%, and 2%) for class-switched isotypes. In all cases, the proportion of observed cells in each cluster fluctuates over time. As expected, the presence or absence of observed phenotypic heterogeneity (in this example, isotype polyclonality) in a cell population is a complex function of relative frequency, fitness, sampling (i.e., bottlenecks), stochasticity, and time (*Ewens, 2012*; *Nowak, 2006*).

## Discussion

### Ig isotype heterogeneity in LCLs

LCL clonality is known to change over time, although the factors involved in this evolution are not fully characterized (*Ryan et al., 2006*). PBMC derivation from multiple donors is an obvious source of cellular heterogeneity in the analyzed samples presented herein. B cells from peripheral blood ($\approx$ 5–10% of all lymphocytes) comprise wide ranges of naïve ($\approx$ 50–80%, mean $\approx$ 65%) and memory ($\approx$ 15–45%, mean $\approx$ 30%) cells, with immature/transitional and plasmablasts accounting for smaller proportions ($\approx$ 1–10%, mean $\approx$ 5% and $\approx$ 0.5–4.5%, mean $\approx$ 2%, respectively; *Perez-Andres et al., 2010*). Within the memory cell compartment, proportions of non-switched (IgM) and switched memory (IgA, IgG, and technically, IgE) are also likely donor-specific. The negligible number of IgE$^+$ cells present across the samples can be explained by the isotype's low frequency in the peripheral blood (*He et al., 2017*). A notable limitation of this study is the lack of access to (GM12878 and GM18502) or retention of (LCL_461 and LCL_777) original donor primary B cells and longitudinal sampling, which would have provided direct insights into donor-dependent cellular heterogeneity.

It is evident from LCL 777 B95-8 and LCL 777 M81 samples that inter-donor differences cannot fully explain the observed isotype heterogeneity in LCLs. While it may be tempting to attribute the observed differences to infection with different viral strains, there is ample experimental evidence that EBV infection does not induce class-switching (*Miyawaki et al., 1991*). The disparity in isotype frequencies is notable since these samples were transformed, cultured, prepared, and sequenced in parallel (i.e., under equivalent conditions and within the same interval).

The polyclonality exhibited within LCL 777 B95-8 and LCL 777 M81 contrast with the dominance of a single isotype in LCL 461 B95-8 and GM18502 samples (in each case, IgG). The only notable difference between LCL 461 B95-8 and LCL 777 B95-8 is that the former sample was in culture substantially longer prior to single-cell library preparation. Given that the GM18502 line was derived more than a decade ago, these observations implicate the influence of culture period in significantly altering the isotype proportions present within LCLs, which is altogether consistent with known (and profound) challenges associated with cell culture (*Hughes et al., 2007*; *Briske-Anderson et al., 1997*; *O'Driscoll et al., 2006*). In this regard, the data from GM12878 merit remark. The finding of polyclonality in this sample is surprising, given that GM12878 has been in culture over a timescale comparable to GM18502 (*Anders and Huber, 2010*). Forgoing the possibility of errors in sample handling or procurement, the persistence of genetic heterogeneity in this line is both intriguing and potentially confounding. Whether or to what extent cellular diversity may influence observed results will inevitably vary on a study-specific basis, but sample-intrinsic variance should be considered even when homogeneity is presumed (*Choy et al., 2008*; *Morley et al., 2004*).

Multiple isotypes within an LCL sample guarantee clonal diversity, but the presence of a single isotype does not necessarily ensure the inverse (intra-sample homogeneity). While not in the scope of the present study, B cell receptor (BCR) 5' single cell sequencing of LCL samples could provide

insights into variable regions and whether subpopulations of a given isotype are the progeny of one or multiple founder cells (and whether this changes over time).

## Viral origins of LCL phenotypic variance

NFκB pathway signaling is constitutively activated by viral LMP-1 in EBV-transformed B cells (*Devergne et al., 1996*). LMP-1 induction of the NFκB pathway is necessary for LCL survival (*Cahir-McFarland et al., 2000*; *Kaye et al., 1993*; *Dirmeier et al., 2003*); however, the observed intra- and inter-LCL variance in transcript levels of NFκB and several of its transcriptional targets add nuance to this picture. Similar profiles of NFκB pathway transcript levels across samples may constitute a snapshot of the most probable distribution arising from stochastic NFκB target expression induced by EBV infection. This may arise from a transcriptional bursting mechanism in which mRNA transcript levels in each cell fluctuate over time (as a Poisson process) while the proportion of cells containing $n$ transcripts in a population at any given time is roughly constant (*Raj et al., 2006*; *Raj and van Oudenaarden, 2008*; *Weinberger et al., 2005*; *Behar and Hoffmann, 2010*; *O'Dea et al., 2007*). Alternatively, or perhaps additionally, variation in NFκB pathway activity may be a manifestation of the different viral latency states present within each sample, as indicated by correlation with host markers of latency IIb and III.

The distinct anticorrelation between NFκB/viral latency program and B lymphocyte differentiation genes is noteworthy. While a mechanism imparting causality to this relationship is not yet fully clear, recent time-resolved bulk transcriptomic data revealed that EBV-induced plasma cell phenotypes (including upregulation of *XBP1*) developed as early as the pre-latent phase of infection (1–14 days; *Mrozek-Gorska et al., 2019*). Correlated expression of *MZB1* with *XBP1*, *TNFRSF17*, *CD27*, and *CD38* support the model that the development of plasma cell characteristics is reminiscent of germinal center differentiation. Single-cell data adds complexity to this finding and its consequences for LCL heterogeneity even after long-term outgrowth. Specifically, EBV transformation in vitro appears to maintain B cells along a continuum of differentiation states, each with varying degrees of similarity to phenotypes observed in vivo (*Price and Luftig, 2015*). In the case of LCL generation, the multiple transcriptional programs of the transformant likely constitute an inescapable source of phenotypic heterogeneity.

The low number of observed lytic cells is likely a consequence of EBV's predominant latency and the fact that lytic reactivation is by nature somewhat incompatible with single-cell RNA-seq methods. However, these small subpopulations provide an interesting case for examination. The spatial proximity of lytic clusters in LCL 777 B95-8 and LCL 777 M81 to plasma-like clusters resulting from tSNE dimensional reduction implied phenotypic similarity; however, we found that this is likely an artifact of the tSNE algorithm since UMAP dimensional reduction did not preserve this proximity. Notwithstanding, *XBP1* upregulation in plasma cells has been shown to transactivate the viral *BZLF1* promoter and induce lytic reactivation (*Sun and Thorley-Lawson, 2007*; *Laichalk and Thorley-Lawson, 2005*). Lytic cells also display relatively high and polyclonal Ig heavy chain expression in addition to other shared characteristics with plasma-like cells (reduced expression of NFκB subunits and its targets). By contrast, lytic cells exhibit notably reduced levels of B cell differentiation transcripts. Thus, viral transcription changes in dynamic response to host cell programs (and vice versa) contribute to the observed LCL diversity. Prior work has shown that the viral proteins EBNA3A and EBNA3C suppress plasma-like phenotypes during EBV latency establishment (*Styles et al., 2017*). The possibility that EBV may undergo lytic reactivation in response to plasma cell differentiation as a means of maintaining persistent latent infection is a topic of future interest.

Host genes upregulated within lytic cluster cells (e.g., *NFATC1*, *MIER2*, *SFN*, and *SGK1*) represent a limited subset of transcription factors associated with B (and T) lymphocyte activation (*Peng et al., 2001*; *Tsitsikov et al., 2001*), several of which have been recently identified at various degrees of enrichment within lytic cells (*Frey et al., 2020*). The presence of *NFATC1* is particularly notable considering the recent report of this factor contributing to the spontaneous lytic phenotype of type 2 EBV by upregulating expression of BZLF1 to promote the lytic gene expression cascade (*Romero-Masters et al., 2020*).

Although PC loadings reveal substantial upregulation of more than a dozen EBV lytic genes, cells within the lytic clusters curiously lack expression of *BZLF1*, which plays a role in the latent-to-lytic transition (*Bhende et al., 2004*). The absence of *BZLF1* reads (and low mRNA counts generally) ostensibly may result from factors including naturally low transcript abundance, reduced transcript

capture efficiency, and/or reduced efficiency of reverse transcription to cDNA owing to RNA secondary structural motifs (*Ozsolak and Milos, 2011*).

## 'Marker-less' subpopulations

The small populations of cells in LCL 461 B95-8, GM12878, and GM18502 characterized by low mitochondrial gene expression and a dearth of canonical B cell markers are curiosities. These cells share similarities with exhausted plasma cells, most notably an apparent loss of Ig heavy chain expression while retaining moderate kappa and light chain expression (*Köhler, 1980*; *Haas and Wabl, 1984*), and hallmarks of oxidative stress including upregulated thioredoxin expression (*Fernando et al., 1992*; *Lu and Holmgren, 2014*; *Muri et al., 2018*; *Muri et al., 2020*). Low levels of NFκB pathway transcripts in these clusters most closely resemble expression profiles of cells with a plasma-like phenotype in the same samples. It is unlikely that these cells are immature, naïve, or transitional B cells, given that neither *IGHM* nor *IGHD* expression is observed. Loss of lineage marker expression is suggestive of a tumor-like phenotype (*Schwering et al., 2003*).

## Factors in the evolution of subclonal heterogeneity

Cellular diversity abounds even within presumptive clonal lines. For LCLs generated from EBV-transformed primary B cells, the list of parameters affecting the cell population's phenotypic profile includes donor-specific frequencies of non-switched and switched memory B cells, heterogeneous states of viral infection, phenotype-specific differential fitness in culture, stochasticity, and time. By definition, some degree of differential fitness exists among cells in each sample as a consequence of the variability in pro-survival, proliferation, and anti-apoptotic genes. Mechanistically, a portion of this variance is expected to arise from heritable yet transient epigenetic signatures (*Shaffer et al., 2020*). Indeed, epigenetic diversity affecting chromatic architecture across LCL subclones from a single donor was recently demonstrated through ChIP-Seq analysis (*Ozgyin et al., 2019*). Lastly, as a principle of evolution, phenotypic differences do not necessarily have to be selected directly; they may simply be carried over in cells possessing other selected features. With respect to the stochastic model presented herein, the simulated phenotype advantage of class-switched memory vs. non-switched memory cells need not be construed as originating from heavy chain isotype expression.

Experimental procedures including cell passaging and the initial transformation itself may contribute to variance among LCLs. As an illustration, consider that 1 million PBMC has around 25,000 B cells, of which 7500 (30%) on average are memory cells of various classes. If the rate of transformation leading to LCL outgrowth is 10%, then ≈750 memory cells out of 1 million PBMCs define the initial isotype frequency of the eventual LCL. This sample size is small relative to the donor's total memory B cell compartment and may lead to founder cell effects. Consequently, B cell population undersampling may be a foregone conclusion in the context of LCL preparation.

Additional studies that utilize time-resolved single-cell sampling from original B cells through early infection and long-term LCL outgrowth in culture will be essential to explore further the factors contributing to longitudinal stability and variation in transcriptional profiles of B cells immortalized by EBV infection. Moreover, while the transcriptomic profiles we report provide a valuable resource, additional molecular layers must be interrogated through parallel -omics techniques (e.g., ATAC-seq and DNA methylation) across individual cells to understand deeply the mechanistic underpinnings of transcriptional heterogeneity.

## Conclusion

Single-cell RNA sequencing reveals that LCLs including widely used commercial lines exhibit substantial phenotypic diversity. During the early stages of LCL generation, EBV infection drives cell proliferation by mimicking the process of B cell activation. After successful LCL outgrowth, infected B cells occupy a range of phenotypic states along a continuum between activation and plasma cell differentiation and, in some cases, exhibit signs of lytic reactivation. The diversity observed within LCLs (and cultured lines generally) can originate from intrinsic heterogeneity within primary cells, transcriptional programs of the viral transformant, and the realization of inherently stochastic processes (including certain gene expression programs) over time. The data reported herein enable extensive hypothesis generation and interrogation of aspects of B cell biology, EBV pathogenesis, and host–virus

interactions. Moreover, this work highlights the importance of considering the possible sources and experimental consequences of cell population heterogeneity when using cultured cell lines.

# Materials and methods

## Key resources table

| Reagent type (species) or resource | Designation | Source or reference | Identifiers | Additional information |
|---|---|---|---|---|
| Biological sample (*Homo sapiens*) | Whole blood | Gulf Coast Regional Blood Center | | Multiple donors; sources of PBMCs for LCL_461 and LCL_777 preparation |
| Cell line (*Homo sapiens*) | B95-8 Z-HT | This paper; *Price et al., 2017* | | Stimulated to obtain B95-8 strain (Type 1 EBV) viral supernatants |
| Cell line (*Homo sapiens*) | M81 | This paper; *Tsai et al., 2013* | | Stimulated to obtain M81 strain (Type 1 EBV) viral supernatants |
| Cell line (*Homo sapiens*) | LCL_461 | This paper; *Price et al., 2017* | | Prepared from donor PBMCs |
| Cell line (*Homo sapiens*) | LCL_777 | This paper | | Prepared from donor PBMCs |
| Cell line (*Homo sapiens*) | GM12878 | Coriell Institute | RRID:CVCL_7526 | White female donor |
| Cell line (*Homo sapiens*) | GM18502 | Coriell Institute | RRID:CVCL_P459 | Yoruba female donor |
| Antibody | Anti-human CD54 (ICAM-1), PE-conjugated (mouse monoclonal) | Biolegend | Cat #353106 | Clone #HA58 |
| Sequence-based reagent | 5' L-VH 1 | This paper; *Tiller et al., 2008* | PCR primers | ACAGG TGCCCAC TCCCAGG TGCAG |
| Sequence-based reagent | 5' L-VH 3 | This paper; *Tiller et al., 2008* | PCR primers | AAGGTG TCCAGTGTGA TGTGCAG |
| Sequence-based reagent | 5' L-VH 4/6 | This paper; *Tiller et al., 2008* | PCR primers | CCCAGATGGG TCCTG TCCCAGG TGCAG |
| Sequence-based reagent | 5' L-VH 5 | This paper; *Tiller et al., 2008* | PCR primers | CAAGGAGTC TGTTCCGAGG TGCAG |
| Sequence-based reagent | 5' L-Vκ 1/2 | This paper; *Tiller et al., 2008* | PCR primers | ATGAGGA TCCCTGC TCAGCTGC TGG |
| Sequence-based reagent | 5' L-Vκ 3 | This paper; *Tiller et al., 2008* | PCR primers | CTCTTCCTCC TGCTACTC TGGCTCCCAG |

*Continued on next page*

*Continued*

| Reagent type (species) or resource | Designation | Source or reference | Identifiers | Additional information |
|---|---|---|---|---|
| Sequence-based reagent | 5′ L-Vκ 4 | This paper; *Tiller et al., 2008* | PCR primers | ATTTCTCTG TTGCTCTGGA TCTCTG |
| Sequence-based reagent | 5′ L-Vλ 1 | This paper; *Tiller et al., 2008* | PCR primers | GGTCC TGGGCCCAG TCTGTGCTG |
| Sequence-based reagent | 5′ L-Vλ 2 | This paper; *Tiller et al., 2008* | PCR primers | GGTCC TGGGCCCAG TCTGCCCTG |
| Sequence-based reagent | 5′ L-Vλ 3 | This paper; *Tiller et al., 2008* | PCR primers | GCTCTG TGACCTCCTA TGAGCTG |
| Sequence-based reagent | 5′ L-Vλ 4/5 | This paper; *Tiller et al., 2008* | PCR primers | GGTCTCTC TCACAGCTTG TGCTG |
| Sequence-based reagent | 5′ L-Vλ 6 | This paper; *Tiller et al., 2008* | PCR primers | GTTC TTGGGCCAA TTTTATGCTG |
| Sequence-based reagent | 5′ L-Vλ 7 | This paper; *Tiller et al., 2008* | PCR primers | GGTCCAATTC TCAGGCTG TGGTG |
| Sequence-based reagent | 5′ L-Vλ 8 | This paper; *Tiller et al., 2008* | PCR primers | GAGTGGATTC TCAGACTG TGGTG |
| Sequence-based reagent | 3′ Cγ CH1 (IgG) | This paper; *Tiller et al., 2008* | PCR primers | GGAAGGTG TGCACGCCGC TGGTC |
| Sequence-based reagent | 3′ Cμ CH1 (IgM) | This paper; *Tiller et al., 2008* | PCR primers | GGGAATTC |
| | TCACAGGAGACGA | Sequence-based reagent | 3′Cα CH1 (IgA) | This paper; *Tiller et al., 2008* |
| PCR primers | TGGGAAGTTTC TGGCGGTCACG | | | |
| Sequence-based reagent | 3′ Cκ 543 (Kappa Light Chain) | This paper; *Tiller et al., 2008* | PCR primers | GTTTCTCGTAG TCTGCTTTGC TCA |
| Sequence-based reagent | 3′ Cλ (Lambda Light Chain) | This paper; *Tiller et al., 2008* | PCR primers | CACCAGTG TGGCCTTG TTGGCTTG |
| Commercial assay or kit | SV96 Total RNA Isolation Kit | Promega | Cat #Z3500 | |
| Commercial assay or kit | High-Capacity cDNA Reverse Transcription Kit | Thermo | Cat #4368814 | |
| Commercial assay or kit | Single Cell 3′ Reagent Kit Protocol, v2 chemistry | 10× Genomics | Cat #CG00052 | |
| Commercial assay or kit | iMag Negative Isolation Kit | BD Biosciences | Cat #558007 | CD19+ B cell isolation from PBMCs |
| Software, algorithm | CellRanger | 10× Genomics | v.2.0.0 | |
| Software, algorithm | Seurat (R package) | *Satija et al., 2015*; *Stuart et al., 2019* | v.3.1.5 | |

## PBMC isolation and transformation with EBV

Whole blood samples from two normal donors (777 and 461) were obtained from the Gulf Coast Regional Blood Center. PBMCs were isolated from each sample by Ficoll gradient (Sigma, # H8889). CD19+ B cells were extracted from each PBMC sample through magnetic separation (BD iMag Negative Isolation Kit, BD, # 558007). Purified B cells were cultured in RPMI 1640 media supplemented with 15% fetal calf serum (FCS, vol./vol., Corning), 2 mM L-glutamine, penicillin (100 units/mL), streptomycin (100 µg/mL, Invitrogen), and cyclosporine A (0.5 µg/mL).

B95-8 and M81 strains of EBV were generated from the B95-8 Z-HT and M81 cell lines, respectively, as described previously (*Johannsen et al., 2004*). Separate bulk infections of B cells were performed by incubating donor B cells with B95-8 Z-HT or M81 supernatants for 1 hr at 37°C, 5% $CO_2$ to produce the following cultures: 777_B95-8, 777_M81, and 461_B95-8. After virus incubation, cells were rinsed in 1× PBS and resuspended in R15 media. LCL outgrowth was achieved from each of these three samples, resulting in LCL_777_B95-8, LCL_777_M81, and LCL_461_B95-8.

## Cell lines and culture

LCL 777_B95-8, 777_M81, and 461_B95-8 were generated in our laboratory by infection of primary human B cells obtained from the Gulf Coast Regional Blood Center with EBV strains B95-8 and M81. These lines were confirmed to be mycoplasma negative using the Sigma Lookout PCR kit.

All three in-house LCL samples were cultured in supplemented RPMI media as described above, substituting 10% FCS instead of 15% FCS. Prior to single-cell sample preparation, LCL_777_B95-8 and LCL_777_M81 were maintained in culture for approximately 1 month, whereas LCL_461_B95-8 was cultured for longer than 6 months. Immediately prior to single-cell sample preparation, LCLs were resuspended and disaggregated.

## LCL samples and data

LCL_777_B95-8, LCL_777_M81, and LCL_461_B95-8 were created as described above. LCLs GM_12878 and GM_18502 were obtained, prepared, sequenced, and aligned as described by Osorio and colleagues (*Osorio et al., 2019*). Briefly, these samples were obtained from the Coriell Institute for Medical Research, cultured for several days, and then prepared as single-cell GEMs (Gel bead in Emulsions) with the 10× Genomics Chromium system using version 2 chemistry for total RNA. Single-cell sequencing libraries were generated using established 10× Genomics protocols, and sequencing was performed with a Novaseq 6000 (Illumina, San Diego). Unique Molecular Identifier (UMI) count matrices were generated from these samples by using CellRanger v.2.1.0 with alignment to the hg38 version of the human reference genome. Additional information about the experimental handling and acquisition of data for GM12878 and GM18502 is provided in the original reference (*Osorio et al., 2019*). Gene-barcode matrix files for each sample were downloaded from the Gene Expression Omnibus (accession ID: GSE126321) and subsequently analyzed along with data from LCL_777_B95-8, LCL_777_M81, and LCL_461_B95-8 samples, while the LCL_461_B95-8 sample was run in a separate experimental batch.

## Single-cell RNA sample preparation and sequencing

Single-cell RNA samples for LCL_777_B95-8, LCL_777_M81, and LCL_461_B95-8 were prepared using the General Sample Preparation demonstrated protocol from 10× Genomics (10×, Manual Part #CG00053) adapted from the original published methods (*Zheng et al., 2017*). Briefly, disaggregated LCLs were resuspended in fresh 1× PBS supplemented with 0.04% BSA, stained with trypan blue to assess viability, and counted using a hemocytometer for preparation to target concentration.

Single-cell libraries for sequencing were prepared from each sample using the methods described in the 10× Genomics Single Cell 3′ Reagent Kit Protocol (v2 chemistry, Manual Part #CG00052). In brief, GEMs were prepared using the 10× Chromium Controller, after which cDNA synthesis and feature barcoding were performed and sequencing libraries for each sample were constructed. Sequencing runs were performed on an Illumina HiSeq 3000/4000 (Illumina, San Diego). Samples for LCL_777_B95-8 and LCL_777_M81 were sequenced in a pooled run in a single HiSeq lane.

Raw base call files (*bcl.gz) from sequencing runs were processed using CellRanger v.2.0.0 to generate fastq files (*fastq.gz) via CellRanger's 'mkfastq' command. CellRanger's 'count' command was then used to align reads from the three in-house LCL samples to the human reference genome (hg38) with the Type 1 EBV reference genome (NC_007605) concatenated as an extra chromosome (reflecting the episomal nature of the EBV genome within infected B cells). This process yielded gene-barcode matrices (UMI count matrices) for subsequent analysis.

## Sample QC, analysis, and visualization

UMI count matrices for all five LCL samples were analyzed using the Seurat single-cell analysis package for R (Seurat v.3.1.5; *Satija et al., 2015*; *Stuart et al., 2019*). Filtered barcode matrices were loaded into Seurat, after which genes present in fewer than three cells and cells expressing fewer than 200 unique RNA molecules (features) and more than 65,000 unique features were filtered out. Additionally, cells in which mitochondrial genes accounted for greater than 5% of all transcripts were excluded from analysis. Beyond the uniform application of QC steps, we did not investigate the potential for batch-specific effects across the five samples run in four experiments. After QC thresholding, feature data were normalized and scored for cell cycle markers. Cell cycle scoring was used to regress out S and G2M gene features to remove variance (and unwanted effects on clustering) in the data sets arising from cell cycle phase. Cell cycle-corrected data were then scaled, and selection was performed to find the highest-variance features. PCA was performed on selected (n = 2000) variable features, and PCs were subsequently used to define distinct subpopulations within each of the five samples. For visualization, PCs were used to generate clusters at various resolutions and dimensionally reduced using tSNE. The R code used to process data and produce figures presented in this manuscript is provided as a supporting file (*Source code 1*), and the Python code used for simulations is provided as a supporting file (*Source code 2*) and is also available on GitHub (https://github.com/esorelle/ig-evo-sim; copy archived at https://archive.softwareheritage.org/swh:1:dir:8c47b2-c0202aa8f255380c742a3cda3ff777abc7/).

## PCR validation experiments

Cell pellets were collected for each of the five LCLs, and total mRNA was extracted from each pellet using the Promega SV96 Total RNA Isolation Kit (Promega, cat # Z3500) and quantified using a NanoDrop 2000 spectrophotometer (Thermo). Total mRNA was then used to create cDNA pools for each sample using a High-Capacity cDNA Reverse Transcription Kit (Thermo, cat # 4368814). Previously reported primer sequences flanking each heavy (IgM, IgA, and IgG) and light chain (Ig kappa and Ig lambda) gene of interest (*Tiller et al., 2008*; listed below) were purchased from Integrated DNA Technologies (IDT) and used to amplify each cDNA pool using standard procedures across a temperature gradient. PCR products and loading dye (Gel Loading Dye, Purple [6×], NEB, cat # B7025) were run on 2% agarose gels with SYBR Safe at 120 V for 45 min with a 100 base pair ladder (NEB, N3231S) and subsequently visualized using a LI-COR Odyssey Fc Imaging System (LI-COR Biosciences). For LCL_777_B95-8 and GM12878, PCR products were sequenced (GeneWiz) and aligned to assess clonality (imgt.org).

## ICAM-1 cell sorting, proliferation, and metabolic assays

LCLs were stained with CD54 (ICAM-1) antibody (PE, Biolegend #HA58) according to the supplier's manual. Then, cells were sorted on a Beckman Coulter Astrios cell sorter by anti-CD54 fluorescence, with ICAM-1-high and ICAM-1-low being defined as the top 15% and bottom 15%, respectively. 24 hr after sorting of ICAM-1-high and ICAM-1-low LCLs, extracellular acidification rate (ECAR) and oxygen consumption rate (OCR) were measured using the Seahorse XF24 extracellular flux analyzer (Agilent Technologies) Cell Energy Phenotype Test. Suspension LCLs were attached to culture plates by using Cell-Tak (BD Bioscience). ECAR and OCR were measured in Seahorse XF Base Medium supplemented with 1 mM pyruvate, 2 mM glutamine, and 10 mM glucose (Sigma Aldrich). ECAR and OCR values were normalized to cell number. For stress measurements, ECAR and OCR were measured over time after injection of oligomycin and FCCP. Metabolic potential measures the ability of cells to meet energetic demands under conditions of stress and is the percentage increase of stressed over baseline ECAR or OCR.

## Stochastic simulations

The concept of a discrete-time Markov chain was adapted to simulate the evolution of phenotype frequencies, using immunoglobulin heavy chain isotype distributions within LCLs as an example. Briefly, the simulation takes as input a cell population of size n comprising B cells of different Ig heavy chain isotype classes at user-defined initial frequencies, fixed probabilities of proliferation in synchronous rounds of cell division, and a constant cell death rate assumption (also user-defined). Within the scope of computational feasibility, users can specify the number of rounds of cell division to simulate and the number of simulation trials to run. Additionally, users may choose to generate simulated cluster data modeled from distinct 2D normal distributions for each isotype for a specified number of trials at fixed intervals (i.e., every $n^{th}$ cell division round). The simulation was implemented in Python, and the code used to generate the simulated data is provided as a supporting file. The code is also available at (add as public GitHub repo) and may be freely implemented and modified.

## Source data files

Raw sequencing data for the three previously unpublished samples (LCL_777_B95-8, LCL_777_M81, and LCL_461_B95-8) are deposited in the NCBI Sequence Read Archive (SRA) and can be accessed along with processed data from the NCBI Gene Expression Omnibus (GEO, Series Accession: GSE158275).

## Acknowledgements

We would like to acknowledge the assistance of the Duke Molecular Physiology Institute Molecular Genomics core (Karen Abramson), the Duke Flow Cytometry Shared Resource (Mike Cook, Lynne and Nancy Martin, and Lynn Martinek), and the Duke School of Medicine Cellular Metabolism Analysis Core Facility (Nancie MacIver and Amanda Nichols) for the generation of key data in this manuscript. We also wish to thank Drs. Eric Johannsen and JJ Miranda for thoughtful discussion and feedback on this work.

## Additional information

### Competing interests

Joanne Dai: Joanne Dai is affiliated with Amgen Inc, The author has no financial interests to declare. The other authors declare that no competing interests exist.

### Funding

| Funder | Grant reference number | Author |
|---|---|---|
| National Institute of Dental and Craniofacial Research | R01-DE025994 | Micah A Luftig |
| National Cancer Institute | T32-CA009111 | Elliott D SoRelle<br>Joanne Dai<br>Micah A Luftig |

The funders had no role in study design, data collection and interpretation, or the decision to submit the work for publication.

### Author contributions

Elliott D SoRelle, Conceptualization, Data curation, Software, Formal analysis, Funding acquisition, Validation, Investigation, Visualization, Writing - original draft, Writing - review and editing, EDS performed Seurat analysis of single-cell sequencing datasets for the five LCLs, developed clonal evolution simulations, helped design validation experiments, prepared figures, and wrote the manuscript; Joanne Dai, Conceptualization, Data curation, Formal analysis, Validation, Investigation, Methodology, Writing - review and editing, JD performed the initial cell isolation (461_B95-8, 777_B95-8, and 777_M81) for the 10X platform as well as the initial CellRanger and Seurat analysis of those three LCLs; Emmanuela N Bonglack, Data curation, Formal analysis, Validation, Investigation, Writing -

review and editing, ENB performed ICAM-1 sorting experiments, growth curves, and Seahorse experiments to assess metabolic differences; Emma M Heckenberg, Data curation, Formal analysis, Investigation, Methodology, Writing - review and editing, EMH performed IgH and IgL RT-PCR and sequencing analysis of LCLs; Jeffrey Y Zhou, Data curation, Formal analysis, Supervision, Validation, Investigation, Writing - review and editing, JYZ performed an independent analysis of the three scRNA-seq datasets derived in house providing important initial insight into the heterogeneity and quality of these samples; Stephanie N Giamberardino, Data curation, Formal analysis, Supervision, Investigation, Methodology, Writing - review and editing, SG performed initial CellRanger and Seurat analysis and training of JD in these suites towards our initial analysis of the in-house derived LCL scRNA-seq data sets; Jeffrey A Bailey, Supervision, Writing - review and editing, JAB provided oversight and guidance to JYZ in the independent analysis of LCL scRNA-seq data of the 461 and 777 LCLs; Simon G Gregory, Conceptualization, Formal analysis, Supervision, Funding acquisition, Methodology, Writing - review and editing, SGG provided oversight of SG and guidance to JD and EDS in developing the analysis pipeline for all scRNA-seq data; Cliburn Chan, Data curation, Formal analysis, Supervision, Investigation, Writing - review and editing, CC supervised EDS along with MAL and provided oversight and review of scRNA-seq analysis as well as the development of the cell proliferation simulations; Micah A Luftig, Conceptualization, Formal analysis, Supervision, Funding acquisition, Investigation, Writing - review and editing, MAL conceived experiments, acquired funding for the project, analyzed data to provided key interpretations regarding the viral/host interactions and activation/differentiation continuum. He provided essential oversight for the conceptual framework of the study and edited the manuscript

## Author ORCIDs
Elliott D SoRelle ![ORCID] https://orcid.org/0000-0002-3362-1028
Joanne Dai ![ORCID] https://orcid.org/0000-0002-9879-4704
Micah A Luftig ![ORCID] https://orcid.org/0000-0002-2964-1907

## Decision letter and Author response
Decision letter https://doi.org/10.7554/eLife.62586.sa1
Author response https://doi.org/10.7554/eLife.62586.sa2

# Additional files

## Supplementary files
- Source code 1. R code used for UMI count matrix processing, analysis, and figure generation.
- Source code 2. Python code used for clonal evolution simulations (also available on github at https://github.com/esorelle/ig-evo-sim).
- Transparent reporting form

## Data availability
Raw sequencing data for the three previously unpublished samples (LCL_777_B958, LCL_777_M81, and LCL_461_B958) are deposited in the NCBI Sequence Read Archive (SRA) and can be accessed along with processed data from the NCBI Gene Expression Omnibus (GEO, Series Accession: GSE158275).

The following dataset was generated:

| Author(s) | Year | Dataset title | Dataset URL | Database and Identifier |
|---|---|---|---|---|
| SoRelle ED, Dai J, Zhou JY, Giamberardino SN, Bailey JA, Gregory SG, Chan C, Luftig MA | 2020 | Single-cell characterization of transcriptomic heterogeneity in lymphoblastoid cell lines | https://www.ncbi.nlm.nih.gov/geo/query/acc.cgi?acc=GSE158275 | NCBI Gene Expression Omnibus, GSE158275 |

The following previously published dataset was used:

| Author(s) | Year | Dataset title | Dataset URL | Database and Identifier |
|---|---|---|---|---|
| Osorio D, Yu X, Yu P, Serepedin E, Cai JJ | 2019 | Single cell RNA sequencing of lymphoblastoid cell lines of European and African ancestries | https://www.ncbi.nlm.nih.gov/geo/query/acc.cgi?acc=GSE126321 | NCBI Gene Expression Omnibus, GSE126321 |

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
