## [Decision Letter]

[Editors’ note: the authors submitted for reconsideration following the decision after peer review. What follows is the decision letter after the first round of review.]

Thank you for submitting your work entitled "Single-cell characterization of transcriptomic heterogeneity in lymphoblastoid cell lines" for consideration by *eLife*. Your article has been reviewed by three peer reviewers, and the evaluation has been overseen by a Reviewing Editor and a Senior Editor. The following individual involved in review of your submission has agreed to reveal their identity: Erik K Flemington (Reviewer #2).

Our decision has been reached after consultation between the reviewers. Based on these discussions and the individual reviews below, we regret to inform you that your work will not be considered further for publication in *eLife*.

The reviewers clearly appreciated your work which, by including five LCL cell lines from different donors and two different virus strains, generates a valuable resource to understand the dynamics between latent and lytic infection in EBV-infected B cells and B cell differentiation versus activation. However, the results and conclusions were not really validated and the work remains quite descriptive. We came to the conclusion, especially taking into regard the comments of reviewer #3, that this work in its current state is not of profound general interest to the readers in the non-EBV field and therefore not suited for publication in *eLife*.

*Reviewer #1:*

The authors report single cell RNA sequencing of five LCL lines. Three of them are recently derived, two with the B95-8 and one with the M81 EBV, and two have been in culture for prolonged periods of time. Two LCLs are from the same donor, one B95-8 and one M81 transformed. Three are predominated by IgG switched memory, one by IgA switched memory and only one by IgM, originating from naïve or unswitched memory B cells. Interestingly, B cell activation and NF-kappaB expression correlates inversely with both antibody isotype class switching and plasma cell differentiation. Lytic EBV gene expression can only be observed in a small subset, primarily in the LCLs from the same donor transformed with B95-8 and M81 viruses but is not higher in the M81 transformed LCL. Finally, the authors model how heterogenous LCL composition might evolve, but the GM12878 cell line demonstrates that this does not necessarily drive to clonality in all instances. The authors suggest that LCLs go through a founder bottleneck that renders possibly every LCL different and that this should be taken into account in using these cellular models.

The study described the comprehensive analysis of five LCLs and generates a valuable resource to understand the dynamics in EBV infected B cells between latent and lytic infection and B cell differentiation versus activation. However, some more information on the correlation of LCL proliferation with gene expression, latent EBV gene expression and antibody isotype composition from uninfected to LCL to lytic reactivation should be provided.

1) The gene expression analysis suggests inverse correlation of B cell activation and differentiation. Is this reflected by LCL proliferation in vitro? Do the LCLs with higher frequencies of differentiated LCLs proliferate slower than for example LCL777 B95-8.

2) The authors report lytic EBV gene expression, but presumably latent transcripts were rarely sequenced due to their low transcript number per cell. Nevertheless, it would be interesting if LMP1 expression frequency is elevated in LCLs with higher frequencies of activated cells and diminished in IgA or IgG expressing cells. The authors should attempt to address these questions by alternative means like flow cytometry or immune fluorescence microscopy.

3) Does lytic EBV reactivation occur in all antibody isotype carrying subpopulations similarly, or is it enriched in XBP1 positive IgG and IgA carrying B cells? Is there any preference between IgA and IgG expressing differentiated B cells?

4) Do the LCLs reflect peripheral B cell composition of the donors at all? Does for example donor 777 have a higher percentage of IgA positive B cells in the peripheral blood than 461.

5) GM12878 might argue that LCL composition could be stable over time in some LCLs and not necessarily drift towards monoclonality. Do the authors have any longitudinal information on BCR isotype composition in their investigated LCLs?

*Reviewer #2:*

This paper demonstrates fairly wide diversity of cell transcriptomes within EBV derived LCL populations. This intra-cell population diversity extends even to cell lines that have been in culture for many years. This likely speaks to the principles driving transcriptional activation which derives from chance intermolecular interactions that are albeit favored or disfavored based on changes in chromatin and chromatin domain structures. The relevance is that the certain level of randomness of these principles can lead to dynamic changes in entire transcription and differentiation programs even within a single cell population. While there is already evidence for these kinds of issues in cell populations, this work brings out these principles in the context of LCLs/EBV (particularly striking are the findings of the presence of plasma blast-like populations and marker-less subpopulations. Overall, this paper provides important insights into the transcriptional and phenotypic diversity that exists in what might have previously been perceived as mostly uniform cell populations of tissue culture LCLs.

The authors have also been able to identify unique transcriptome signatures for reactivating cells which is potentially interesting from the standpoint of informing us on the nature of apparently stochastic events that trigger this transition to the EBV lytic phase. Nevertheless, given the lack of detection of BZLF1 (and possibly other lytic transcripts?), it would be helpful if the authors could provide additional evidence that these are true lytic cells vs abortive lytic cells (the latter of which would itself would be an interesting finding). It would be helpful if the authors could plot distributions of the percentage of viral lytic transcript reads to cell transcript reads in these populations of cells (this is hard to gauge from Figure 3C). Since herpesviral lytic infection typically results in a substantial proportion of lytic transcripts (minimum of 10% of all reads), this would help determine whether most of these cells are truly lytic or abortive lytic (perhaps through some epigenetic changes that lead to a higher level of transcriptional bursting of the EBV genome).

*Reviewer #3:*

Summary: Lymphoblastoid Cell Lines (LCLs) are induced by infecting primary B cells with Epstein-Barr Virus (EBV) and constitute a widely used cell line model in molecular biology, oncology and immunology. Understanding the intra- and inter-heterogeneity of these cell lines using single-cell analysis is key to design research and interpret experimental results. The authors induced three cell lines using two EBV strains and they performed single cell RNA-seq using the 10x platform and conducted a very classical analysis using Seurat. They added two datasets from the literature (Osorio et al., 2019). They describe that each cell line has a certain level of intra-heterogeneity with different Ig expression patterns, different maturation stages (Figure 1 and 2), as well as exhibit different viral lytic/latent stages (Figure 3) and mitochondrial gene expression (Figure 4).

General comments: This study is needed and interesting as a resource for the community of scientists using these cell lines but I see major problems in the design of the study and the analysis. Beyond doing scRNA-seq and displaying clustering analysis, the final aim and the novelty of the study are not clear to me. It is neither clear how one can use such analysis to guide his research. As presented, the data analysis is very preliminary (Figure 1—figure supplement 15-19 and Figure 4—figure supplement 1-4) and the study needs substantial improvements before publication in a top-tier journal such as *eLife*. No functional analysis is provided to indeed demonstrate that intra- and inter-variability has implications in study design.

1) Design of the experiments: the overall scope of the study is very limited with only five different donors and it is not clear what is the contribution of the different viral strains. The rational of the study design is not outlined. The authors have chosen to look at the transcriptome only; we would have expected to have more -omics for a resource paper such as ATAC-seq and methylome to document the underlying heterogeneity of the cells.

2) The description of the variability of LCLs is not novel. Ozgyin et al., 2019 have already provided a genotype-independent functional genomic variability of the LCLs. This study is not quoted.

3) The variability between the five LCLs is not clearly delineated. Each cell line is heterogenous (Figure 1 to 4) but the authors have studied the cell lines independently without attempting to merge the data. The inter-heterogeneity is very poorly documented. The data analysis to perform a merge would require substantial computational work that goes beyond the scope of a two-month revision. It would be very important to explain the contribution of the inter-individual genomic variability between donors.

4) The authors claim in the Abstract that "This heterogeneity is likely attributable to intrinsic variance in primary B cells and host-pathogen dynamics." In a publication in a top journal such as *eLife*, we expect that such a claim is documented with experiments. The authors claim that "primary cell heterogeneity, random sampling, time in culture, and even mild differences in phenotype-specific fitness" can contribute to such heterogeneity. These are very general claims that do not help and are not supported with experiments.

[Editors’ note: further revisions were suggested prior to acceptance, as described below.]

Thank you for submitting your article "Single-cell characterization of transcriptomic heterogeneity in lymphoblastoid cell lines" for consideration by *eLife*. Your article has been reviewed by three peer reviewers, and the evaluation has been overseen by a Reviewing Editor and Päivi Ojala as the Senior Editor. The following individual involved in review of your submission has agreed to reveal their identity: Erik K Flemington (Reviewer #2).

The reviewers have discussed the reviews with one another and the Reviewing Editor has drafted this decision to help you prepare a revised submission.

Summary:

Lymphoblastoid Cell Lines (LCLs) are induced when primary B cells are infected with the human herpesvirus Epstein-Barr Virus (EBV) and are a widely used model in molecular biology, oncology, and immunology. The authors have used single cell transcriptomics to demonstrate substantial phenotypic heterogeneity within and across LCLs. Hence, this work is important for researchers working with LCLs for the design and interpretation of experiments.

Revisions:

Please include in your discussion the points raised by reviewers #1 and #3 about the limitations of this study:

Reviewer #1: Due to lack of donor material prior to transformation by EBV the authors could not assess if the heterogeneity of their cell lines was donor dependent. Furthermore, they could not provide any information on the longitudinal stability of the reported heterogeneity.

Reviewer #3: It remains to uncover the origin of this variability using other layers of -omics and longitudinal sampling of the cells along the transformation process.

---

## [Author Response]

[Editors’ note: The authors appealed the original decision. What follows is the authors’ response to the first round of review.]

The reviewers clearly appreciated your work which, by including five LCL cell lines from different donors and two different virus strains, generates a valuable resource to understand the dynamics between latent and lytic infection in EBV-infected B cells and B cell differentiation versus activation. However, the results and conclusions were not really validated and the work remains quite descriptive. We came to the conclusion, especially taking into regard the comments of reviewer #3, that this work in its current state is not of profound general interest to the readers in the non-EBV field and therefore not suited for publication in eLife.

We thank the reviewers and editors for their consideration and recognition of our work. Based on the unanimous conclusion by the reviewers and editors that the work “generates a valuable resource,” we request that the submission be reconsidered as a Tools and Resources article rather than a Research Article.

We have revised the manuscript with data from three additional experiments that directly address reviewer requests for validation and functional implications of our findings. Further, we have provided responses and references to address other questions raised during the review. These are described further in the point-by-point responses to each review comment. We believe that these revisions have strengthened our manuscript substantially and adequately resolve the reviewers’ concerns.

We strongly believe this work will be of interest to readers outside of the EBV field because of the widespread use of LCLs in genetic and immunological research (as we noted in the manuscript and in our cover letter accompanying the original submission). The diversity in viral transcription observed within the datasets will certainly be relevant to EBV researchers; the effects of viral transformation on host cell phenotypic heterogeneity for these widely used models are likewise relevant to a more general readership.

Reviewer #1:[…]1) The gene expression analysis suggests inverse correlation of B cell activation and differentiation. Is this reflected by LCL proliferation in vitro? Do the LCLs with higher frequencies of differentiated LCLs proliferate slower than for example LCL777 B95-8.

We have provided new data to address this question. We sorted LCLs by ICAM (a marker for activated cells) expression and found that ICAM-high populations exhibit more rapid proliferation than ICAM-low populations in the days immediately after sorting (Figure 2—figure supplement 8 in the revised manuscript). This highlights a functional implication of the observed transcriptomic heterogeneity with respect to cell population growth dynamics.

2) The authors report lytic EBV gene expression, but presumably latent transcripts were rarely sequenced due to their low transcript number per cell. Nevertheless, it would be interesting if LMP1 expression frequency is elevated in LCLs with higher frequencies of activated cells and diminished in IgA or IgG expressing cells. The authors should attempt to address these questions by alternative means like flow cytometry or immune fluorescence microscopy.

As recently reported by our lab (Messinger et al., 2019), FACS sorting by ICAM-1 expression serves as a reliable proxy for viral LMP-1 expression due to tight correlation (via EBV induction of NF-κB pathways) [see Figures 1B-D, 2C in Messinger et al., (2019)] and correlates directly to the extent of DNA replication in LCLs [see Figure 3D]. Additional RNA-FiSH data from the cited paper [see Figure 6] confirms that only a fraction of cells within LCL populations express LMP1. These findings are consistent with the newly provided cell proliferation and metabolic data sorted by ICAM status, which collectively indicate the functional role of elevated LMP1 in activated cell subpopulations.

3) Does lytic EBV reactivation occur in all antibody isotype carrying subpopulations similarly, or is it enriched in XBP1 positive IgG and IgA carrying B cells? Is there any preference between IgA and IgG expressing differentiated B cells?

The limited number of lytic cells preclude a definitive answer to this question; however, we note that cells expressing each of the three heavy chain isotypes are observed within the lytic cluster in LCL 777 B95-8 at isotype frequencies roughly corresponding to those of the whole cell population (Figure 1—figure supplement 5). This suggests similarity in lytic reactivation rates across isotypes [Note: there are even fewer lytic cells in LCL 777 M81, however the presence of IgA- and IgG-expressing cells is likewise observed (Figure 1—figure supplement 6)].

4) Do the LCLs reflect peripheral B cell composition of the donors at all? Does for example donor 777 have a higher percentage of IgA positive B cells in the peripheral blood than 461.

The diversity among LCLs almost certainly reflects different donor peripheral B cell compositions (as well as other factors including passaging and stochasticity – see Figure 5 simulations and corresponding results/discussion). However, we did not retain (and in the case of GM12878 and GM18502, do not have access to) primary cell samples prior to transformation, so it is not possible to answer this question dispositively for the reported lines. As noted by Heath and Rickinson [Heath et al., PLoS Pathogens (2012), https://journals.plos.org/plospathogens/article/comments?id=10.1371/journal.ppat.1002697], we would expect that this does reflect B cell composition of donor, however such composition can change over time.

5) GM12878 might argue that LCL composition could be stable over time in some LCLs and not necessarily drift towards monoclonality. Do the authors have any longitudinal information on BCR isotype composition in their investigated LCLs?

We did not acquire longitudinal data in this work.

Reviewer #2:[…]The authors have also been able to identify unique transcriptome signatures for reactivating cells which is potentially interesting from the standpoint of informing us on the nature of apparently stochastic events that trigger this transition to the EBV lytic phase. Nevertheless, given the lack of detection of BZLF1 (and possibly other lytic transcripts?), it would be helpful if the authors could provide additional evidence that these are true lytic cells vs abortive lytic cells (the latter of which would itself would be an interesting finding). It would be helpful if the authors could plot distributions of the percentage of viral lytic transcript reads to cell transcript reads in these populations of cells (this is hard to gauge from Figure 3C). Since herpesviral lytic infection typically results in a substantial proportion of lytic transcripts (minimum of 10% of all reads), this would help determine whether most of these cells are truly lytic or abortive lytic (perhaps through some epigenetic changes that lead to a higher level of transcriptional bursting of the EBV genome).

Thank you to reviewer #2 for making this important suggestion. We have re-analyzed the level of host and viral gene expression in the lytic cell clusters and have found that the percentage of lytic transcripts in cells varies across the lytic cluster (LCL 777 B95-8, Figure 3—figure supplement 1 in the revised manuscript). In the highest-expressing cells, the 20 most prevalent lytic genes account for 10-15% of all transcripts, which is consistent with truly lytic cells. Interestingly, about half of the cells in the cluster have less than 10% lytic transcript content. As the reviewer has noted, this could indicate abortive lytic replication; alternatively, these could be cells that had only recently initiated lytic replication at the time of sample preparation. We have added this analysis and corresponding discussion to the manuscript.

Reviewer #3:Summary: Lymphoblastoid Cell Lines (LCLs) are induced by infecting primary B cells with Epstein-Barr Virus (EBV) and constitute a widely used cell line model in molecular biology, oncology and immunology. Understanding the intra- and inter-heterogeneity of these cell lines using single-cell analysis is key to design research and interpret experimental results. The authors induced three cell lines using two EBV strains and they performed single cell RNA-seq using the 10x platform and conducted a very classical analysis using Seurat. They added two datasets from the literature (Osorio et al., 2019). They descibe that each cell line has a certain level of intra-heterogeneity with different Ig expression patterns, different maturation stages (Figure 1 and 2), as well as exhibit different viral lytic/latent stages (Figure 3) and mitochondrial gene expression (Figure 4).General comments: This study is needed and interesting as a resource for the community of scientists using these cell lines but I see major problems in the design of the study and the analysis. Beyond doing scRNA-seq and displaying clustering analysis, the final aim and the novelty of the study are not clear to me. It is neither clear how one can use such analysis to guide his research. As presented, the data analysis is very preliminary (Figure 1—figure supplement 15-19 and Figure 4—figure supplement 1-4) and the study needs substantial improvements before publication in a top-tier journal such as eLife. No functional analysis is provided to indeed demonstrate that intra- and inter-variability has implications in study design.

We thank reviewer #3 for acknowledging the utility of our reported work for a broad community of researchers beyond the EBV community. We believe that the newly added data and analysis signify substantial improvements to the manuscript with respect to validation and functional implications for LCLs.

Respectfully, it is unclear why the reviewer cites 9 out of 38 total figures and figure supplements to broadly assert that the presented data is “very preliminary” – we provided these specific supplements (principal component heatmaps and clustering resolution screens) for completeness and analytical transparency (for example, different clustering resolutions may be required depending on the desired analysis). This assertion does not substantively challenge the key data and findings presented in the main figures.

Regarding aim and novelty, we note several key aspects of the work including: 1) the first (to our knowledge) evidence of multiclonality in LCLs at single-cell resolution, 2) single-cell characterization of inverse expression patterns for B cell activation and differentiation programs (now supplemented with additional functional and validation data), 3) the first single-cell dataset revealing EBV lytic replication at appreciable levels, and 4) the first report of LCL subpopulations with distinct metabolic profiles.

With respect to how such analysis might allow one to guide her own research, the datasets clearly identify numerous genes of interest and correlations that can inform (to name just a few areas):

1) Subsequent targeted studies in B cell activation and metabolism; modeling of host-pathogen dynamics

2) Guides to developing molecular reporters and other tools for live-cell studies

3) Transcriptome-wide evaluation of the effects of chemical perturbations including potential therapeutics (for example, the class of drugs that aims to treat viral infection by inducing higher rates of lytic reactivation).

This work clearly opens new avenues of research, including refinement of a model for B cell functional states and dynamics in the context of EBV infection and associated lymphomas.

We provide further detailed addresses to reviewer #3’s concerns below, which we believe have resulted in a substantially improved manuscript.

1) Design of the experiments: the overall scope of the study is very limited with only five different donors and it is not clear what is the contribution of the different viral strains. The rational of the study design is not outlined. The authors have chosen to look at the transcriptome only; we would have expected to have more -omics for a resource paper such as ATAC-seq and methylome to document the underlying heterogeneity of the cells.

A previously uncharacterized inverse relationship between activation and differentiation transcriptional signatures at the single cell level is consistently shown in 5 LCLs from different donors prepared and analyzed in different labs. At the same time, notable and meaningful clonal differences are found to exist within and across the LCLs, demonstrating that nominally identical cell populations should not be assumed to be monolithic with respect to identity or fitness. We cite data linking some of these differences to variation in viral transcriptional states and demonstrate the effects on cell proliferation. In and of themselves, these are valuable findings. While inclusion of additional samples may offer further confirmation or refinement of the generalizability of either finding, it is not immediately clear to us that further samples are necessary in this case. The reviewer contends that five samples constitute a limited study but provides no rationale or statistical basis for what defines a study of sufficient size – this is necessarily dependent on the nature and magnitude of what is being investigated/tested. Because the findings of the study were not known *a priori*, defining such a standard would have proven challenging. Whether 100% of LCLs uniformly exhibit the B cell activation / differentiation continuum or what percentage of LCLs display polyclonality are questions best addressed by future studies.

The contributions from testing different viral strains (B95-8 and M81) in cells from the same donor are articulated throughout the manuscript.

We believe the rationale and aim of the study is clearly defined in the manuscript Introduction: “Although studied extensively, complete characterization of the viral and host determinants of growth arrest versus immortalization of early-infected cells remains elusive.^27^ As one consequence, it is unclear whether or to what extent viral transformation may influence the resulting LCL cell populations. The possibility of significant phenotypic diversity within and across LCL samples warrants consideration, given the intrinsic variance of the human primary B cell repertoire^28,29^ and the multiplicity of viral transcription programs active in the journey to immortalization.”

Reviewer #3 is correct that additional -omics data would add layers of functional detail such as epigenetic profiling to the study. While we appreciate the reviewer’s focus and expertise in multi-omics analyses and the importance of these techniques, this work was conceived and executed with a different scope and focus. We feel that the work in its present form provides a rich resource and compelling findings that will be sufficiently valuable to the readership.

2) The description of the variability of LCLs is not novel. Ozgyin et al., 2019 have already provided a genotype-independent functional genomic variability of the LCLs. This study is not quoted.

We thank reviewer #3 for providing this reference. We have cited and discussed it within our revised manuscript. The statement that “variability of LCLs is not novel” is rather vague and reductive – the work by Ozgyin characterizes certain aspects of variability within LCLs, and our work characterizes aspects of variability that are distinct from that paper’s scope. Below, we highlight several key differences of our manuscript relative to the work by Ozgyin and colleagues:

1) Ozgyin et al. provide a thorough exploration of bulk transcriptomic and epigenetic heterogeneity in multiple LCLs cultured from a single donor. By contrast, we report a conserved continuum of transcriptomic heterogeneity across LCLs from multiple donors.

2) Ozgyin et al. show that heritable epigenetic signatures persist across different samples from the same donor and that these differences may have pharmacological implications. Our work shows that transcriptional differences within LCLs are correlated to different functional states intrinsic to B cell biology and are influenced by heterogeneity in viral programs.

3) Ozgyin et al. present FACS data showing light chain variability across samples from the same donor (a valuable finding). Our work presents single-cell sequencing data showing heavy chain clonality within and across donors, as well as light chain data.

4) Ozgyin et al. focus on host cell transcription as influenced by chromatin accessibility. Our work focuses on the effects of host-pathogen interactions by reporting both host cell and viral reads and their relative expression in individual cells within LCLs.

5) Our work describes a stochastic modeling framework to explore the effects of differential phenotypic fitness and other experimental considerations on LCL evolution.

6) Notably, both works highlight the broad utility of and interest in LCLs across research disciplines

3) The variability between the five LCLs is not clearly delineated. Each cell line is heterogenous (Figure 1 to 4) but the authors have studied the cell lines independently without attempting to merge the data. The inter-heterogeneity is very poorly documented. The data analysis to perform a merge would require substantial computational work that goes beyond the scope of a two-month revision. It would be very important to explain the contribution of the inter-individual genomic variability between donors.

The revised manuscript provides a characterization of the merged datasets (Figure 1—figure supplement 5 in the revised manuscript). It shows that donor-specific differences are a dominant source of LCL heterogeneity. The fact that donor-specific genomic differences dominate relative to conserved intra-sample trends is unsurprising. This analysis reinforced discussion related to the stochastic model of cell phenotype evolution – that the contribution of inter-individual genomic variability is significant.

4) The authors claim in the Abstract that "This heterogeneity is likely attributable to intrinsic variance in primary B cells and host-pathogen dynamics." In a publication in a top journal such as eLife, we expect that such a claim is documented with experiments. The authors claim that "primary cell heterogeneity, random sampling, time in culture, and even mild differences in phenotype-specific fitness" can contribute to such heterogeneity. These are very general claims that do not help and are not supported with experiments.

These are general and foundational principles. In the case of the effects of host-pathogen dynamics, please refer to the substantial body of research describing EBV LMP1-mediated upregulation of NF-κB signaling pathways and the newly-provided data showing higher proliferation of ICAM-1 – high (an LMP1 proxy) cells within LCLs. In the case of the effects of random sampling, time in culture, and phenotype-specific fitness, we were referring to results of stochastic simulations. While not so dispositive as clean results from wet lab experiments, models and simulations can be valuable tools for research and guide subsequent studies.

[Editors’ note: what follows is the authors’ response to the second round of review.]

Revisions:Please include in your discussion the points raised by reviewers #1 and #3 about the limitations of this study:Reviewer #1: Due to lack of donor material prior to transformation by EBV the authors could not assess if the heterogeneity of their cell lines was donor dependent. Furthermore, they could not provide any information on the longitudinal stability of the reported heterogeneity.

We have added the following sentences to the Discussion section to address these points:

“A notable limitation of this study is the lack of access to (GM12878 and GM18502) or retention of (LCL_461 and LCL_777) original donor primary B cells and longitudinal sampling, which would have provided direct insight into donor-dependent cellular heterogeneity.”

“Additional studies that utilize time-resolved single-cell sampling from original B cells through early infection and long-term LCL outgrowth in culture will be essential to explore further the longitudinal stability and variation in transcriptional profiles following EBV infection.”

Reviewer #3: It remains to uncover the origin of this variability using other layers of -omics and longitudinal sampling of the cells along the transformation process.

We have added the following sentences to the Discussion section to address these points:

“Additional studies that utilize time-resolved single-cell sampling from original B cells through early infection and long-term LCL outgrowth in culture will be essential to explore further the longitudinal stability and variation in transcriptional profiles following EBV infection. Moreover, while the transcriptomic profiles we report provide a valuable resource, additional molecular layers must be interrogated through parallel -omics techniques (e.g., ATAC-seq, DNA methylation) across individual cells to understand deeply the mechanistic underpinnings of transcriptional heterogeneity.”